# Recent contrasting behaviour of mountain glaciers across the European High Arctic revealed by ArcticDEM data

Jakub Małecki[1]

[1]Cryosphere Research Unit, Adam Mickiewicz University, Poznań, Poland

*Correspondence to*: Jakub Małecki (malecki.jk@gmail.com)

**Abstract.** Small land-terminating mountain glaciers are a widespread and important element of Arctic ecosystems, influencing local hydrology, microclimate, and ecology. Due to their relatively small ice volumes, this class of ice masses is particularly sensitive to the significant ongoing climate warming in the European sector of the Arctic, i.e. in the Barents Sea area. Archipelagos surrounding the Barents Sea, i.e. Svalbard (SV), Novaya Zemlya (NZ), and Franz Josef Land (FJ), host numerous populations of mountain glaciers, but their response to recent strong warming remains understudied in most locations. This paper aims to obtain a snapshot of their state by utilizing high-resolution elevation data (ArcticDEM) to investigate the recent (ca. 2011-2017) elevation and volume changes of 382 small glaciers across SV, NZ, and FJ. The study concludes that many mountain glacier sites across the Barents Sea have been in a critical imbalance with the recent climate and might melt away within the coming several decades. However, deviations from the general trend exist, e.g. a cluster of small glaciers in north SV has been experiencing thickening. The findings reveal that near-stagnant glaciers might exhibit contrasting behaviours (fast thinning vs. thickening) over relatively short distances, which is a challenge for glacier mass balance models, but also an opportunity to test their reliability.

## Introduction

The Arctic holds nearly a half of the global area and volume of glacier ice outside of ice sheets (Farinotti et al., 2019) and has been warming faster than other regions over the past decades (Screen and Simmonds, 2010). Recent progress in large-scale glacier mass balance studies, utilizing climate modelling, satellite altimetry, gravimetry and photogrammetry, has fundamentally improved our knowledge on the general trends of glacier change in the Arctic and the contribution of its ice masses to the global sea-level rise (e.g. Moholdt et al., 2012; Box et al., 2018; Noël et al., 2018; 2020; Wouters et al., 2019; van Pelt et al., 2019). These studies might have a relatively coarse spatial resolution (≥ 1 km) and, thus, more limited coverage of the smallest (e.g. one kilometre-scale) glaciers. However, a recently published dataset of glacier elevation changes by Hugonnet et al. (2021) delivered data for nearly every glacier in the world at 100 m resolution, but its performance on small mountain glaciers has not yet been compared with independent data.

Small ($< 30$ km$^2$) land-terminating ice masses (hereafter termed 'mountain glaciers') are the most widespread and numerous class of glaciers north of the Arctic circle and are vulnerable to climate warming due to their small ice volumes and limited vertical extents. They commonly exhibit low horizontal and vertical ice velocities, resulting from predominantly cold thermal regimes and shallow ice thickness, so their elevation changes might closely reflect the climate-driven point mass balance, being only little modified by ice dynamics (e.g. Melvold and Hagen, 1998; Nutall and Hodgkins, 2005; Hambrey et al., 2005; Hagen et al., 2005). Therefore, studying elevation changes of Arctic mountain glaciers might not only help predict future fluctuations of their remaining volume but also provide a piece of valuable proxy information about the spatial variability of glacier mass balances and the state of climate over remote areas. This might be useful in developing glacier mass balance models over regions with limited direct measurements for model calibration or validation, e.g. on most Arctic islands. This task is important for broadening our understanding of the possible responses of the cryosphere to climate warming, particularly in areas experiencing very fast changes, e.g. in Arctic Europe.

The European sector of the Arctic surrounding the Barents Sea has been nicknamed 'the Arctic warming hotspot' by Lind et al. (2018) due to the severe climate and oceanic warming and rapid retreat of sea ice it has been experiencing (e.g. Årthun et al., 2012; Comiso and Hall, 2014; Kohnemann et al., 2017). These strong shifts potentially exposed mountain glaciers of the nearby archipelagos to increased melt, i.e. in Svalbard (SV), Novaya Zemlya (NZ) and Franz Josef Land (FJ).

The available evidence from SV, the best studied of these regions, clearly shows that over the past decades the whole archipelago has been losing ice and that mountain glaciers at most sites have been experiencing accelerating thinning at high elevations (e.g. Nuth et al., 2007; Kohler et al., 2007; James et al., 2012; Małecki 2016; van Pelt et al., 2019; Noël et al., 2020; Schuler et al., 2020; Geyman et al., 2022). Recent research covering the Russian sector of the Barents Sea demonstrated that both small and large ice masses of NZ and FJ have been also losing mass at an increasing rate (Ciracì et al., 2018; Zheng et al., 2018; Hugonnet et al., 2021; Sommer et al., 2022).

Although mountain glaciers are likely the first to disappear in a rapidly warming Arctic, detailed information on their recent state (e.g. glacier-wide elevation change rates, volume change rates, relationships between thinning and elevation) have been reported only for smaller locations across the Barents Sea islands. This paper focuses on recent elevation changes among mountain glaciers groupings, mostly understudied, across SV, NZ and FJ over the period ca. 2011-2017. Its objectives are to: (i) define the baseline style of the recent behaviour of mountain glaciers and detect anomalies from the general trend using high-resolution elevation data; (ii) to estimate the future lifespan of mountain glaciers and detect sites where these are critically endangered by climate warming; and (iii) to deliver proxy glacier mass balance data which might serve as a calibration/validation for existing or future glacier mass balance models for remote areas.

## 2 Study area

### 2.1 Climatology of the Barents Sea area

The Barents Sea is strongly influenced by warm Atlantic waters (Fig. 1a), particularly to the west (e.g. West Spitsbergen Current) and south (e.g. North Cape Current), making the climate of these two sectors relatively mild (Figs. 1b and 1c) and wet (Fig. 1d). For this reason, waters west of SV and southwest of NZ are free of sea ice, even during the winter (Fig. 1e). During summer, the Barents Sea is typically ice-free, except for around FJ (Fig. 1f), where the Atlantic influence is reduced, making climate there the coolest and driest in the region.

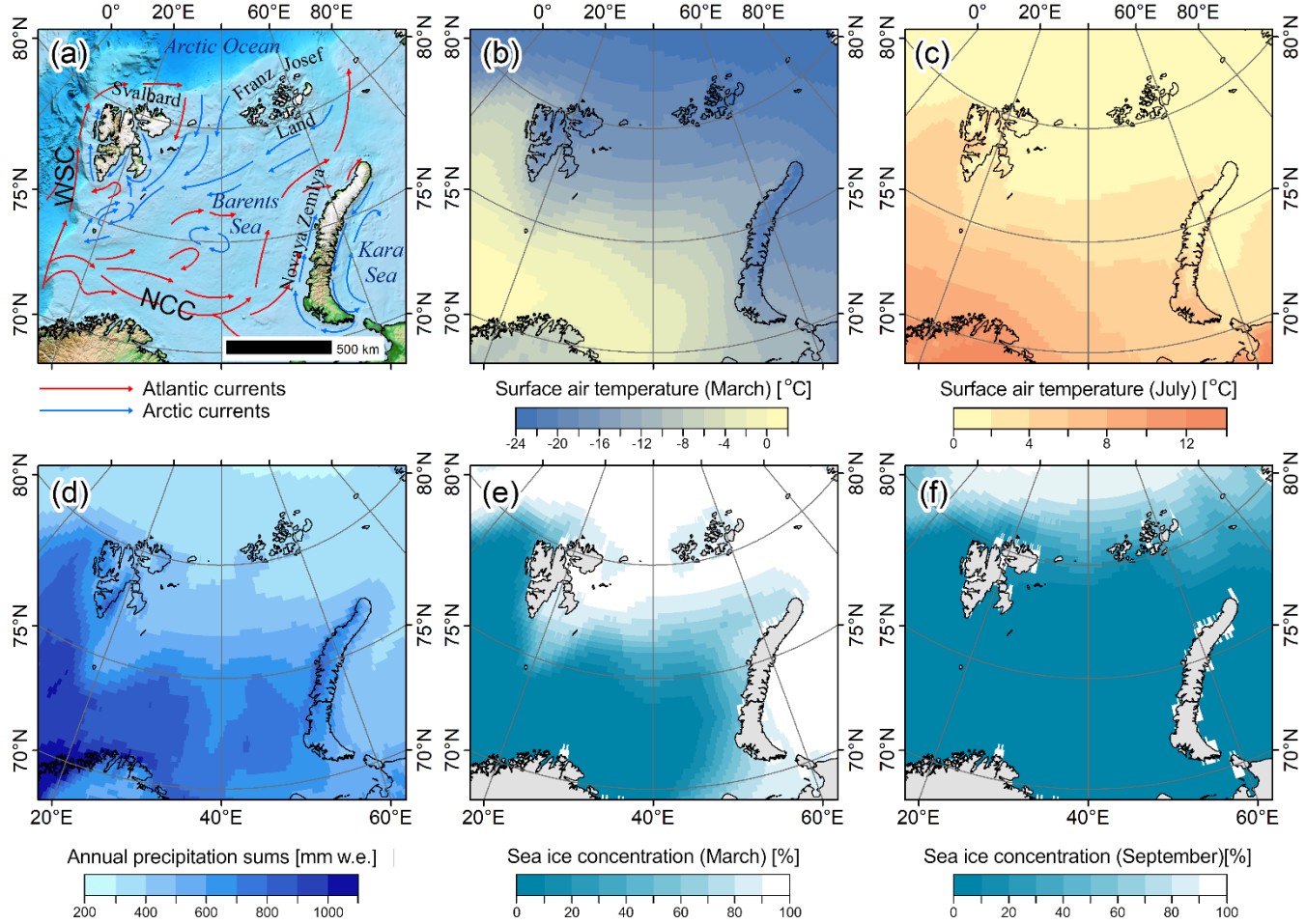

**Figure 1 (a) Topography of the Barents Sea area and transport of Atlantic water (red arrows) and Arctic water (blue arrows) after the Institute of Marine Research in Bergen, Norway. WSC - West Spitsbergen Current; NCC - North Cape Current. (b-f ) ERA5 1981-2010 climatology for the Barents Sea area (b - March near-surface air temperature, c - July near-surface air temperature, d - annual precipitation sums, e - sea ice concentration in March, f - sea ice concentration in September).**

Over the past decades, the Barents Sea area has been experiencing strong ocean and atmospheric changes resulting in water temperature increase and sea ice retreat, the so-called Atlantification (Årthun et al., 2012; Barton et al., 2018; Asbjørnsen et al.,

2020). Instrumental records indicate heat inflow from the Atlantic to the Arctic Ocean via the West Spitsbergen Current has been increasing (Piechura and Walczowski, 2009; Walczowski and Piechura, 2011), contributing to strong sea ice decline over the area (Årthun et al., 2012; Onarheim et al., 2014; Barton et al., 2018). Concurrently, mean sea surface temperature across the
Barents Sea increased over the period 1981-2012 (Comiso and Hall, 2014). A strong positive trend in annual surface air temperature exceeding 1 °C/decade has been reported for some sites, accompanied by an even stronger warming rate exceeding 3 °C/decade during winter months, particularly around SV and NZ (e.g. Isaksen et al., 2016; Kohnemann et al., 2017; Lind et al., 2018; Wawrzyniak and Osuch, 2020; Nordli et al., 2020; Dahlke et al., 2020).

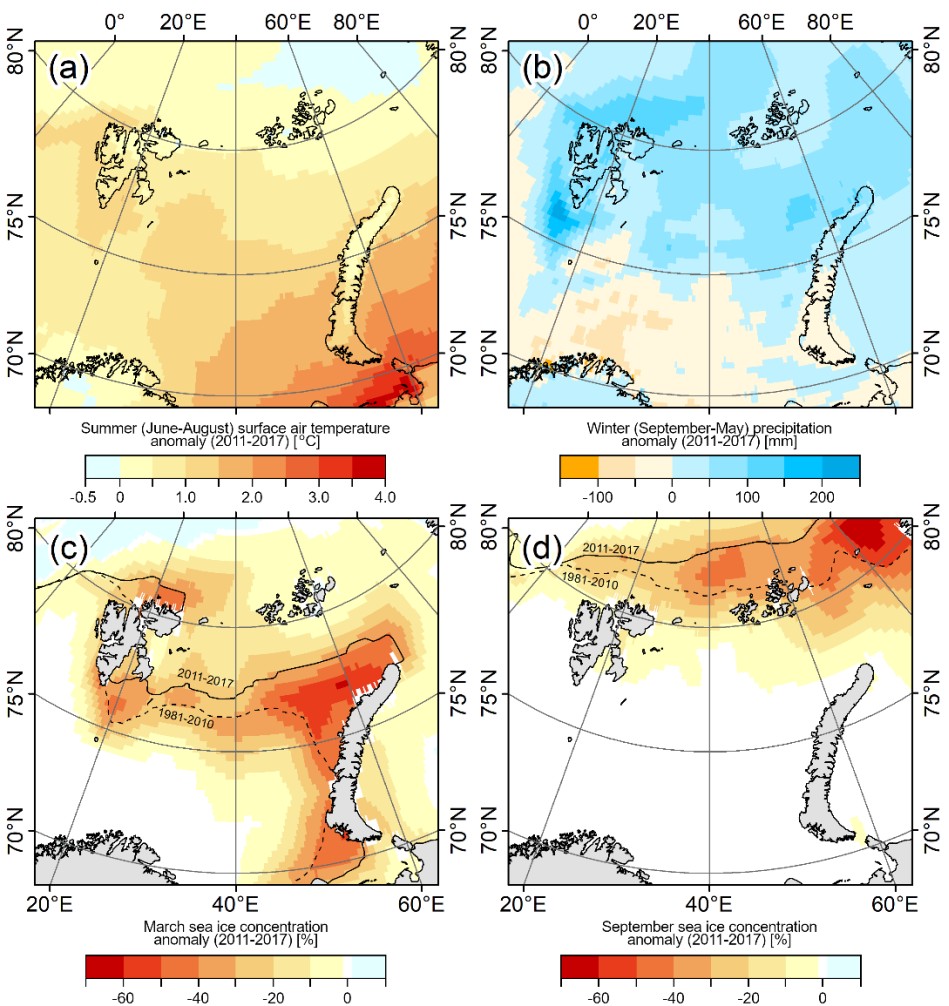

**Figure 2 The 2011-2017 anomalies of selected climate variables from the ERA5 reanalysis in reference to the period 1981-2010. (a) Summer (June-August) near-surface air temperature anomaly. (b) Winter (September-May) precipitation anomaly. (c) March sea ice concentration anomaly. (d) September sea ice concentration anomaly. Dashed and solid black lines in c and d – 50 % sea ice concentration extent for 1981-2010 and 2011-2017, respectively.**

The global climate reanalysis ERA5 dataset (Hersbach et al., 2020) shows that these trends prevail over the period of this study (roughly 2011-2017). Over SV, NZ and FJ summer (June-August) surface air temperatures in the period 2011-2017 were higher by ca. 0.5-1.0°C compared to the reference period 1981-2010 (Fig. 2a). Most land areas also experienced slightly increased winter (September-May) precipitation by ca. 50-100 mm, or ca. 10-20 % (Fig. 2b). In March the sea ice concentration decline was most apparent along the western coast of NZ and around SV (Fig. 2c), and in September, it is most visible to the northeast of FJ (Fig. 2d). The ERA5 reanalysis presented by Morris et al. (2020) for the same period 2011-2017 indicates a strong increase in June-August sea surface temperature (by ca. 2°C; period of reference 2004-2008) around SV and NZ, accompanied by June-August sea ice concentration drop along eastern coasts of the three study regions.

## 2.2 Glaciers of Svalbard, Novaya Zemlya and Franz Josef Land

All three archipelagos of the Barents Sea are heavily glacier covered and mountain glaciers comprise ca. 10 % of their overall ice area. The best-studied region is SV, hosting 34,000 km$^2$ of land ice, mainly on its largest islands: mountainous Spitsbergen and more gentle Nordaustlandet and Edgeøya. According to the Randolph Glacier Inventory (RGI) v6.0 (Pfeffer et al., 2014), approximately 85 % are mountain glaciers, which comprise 14 % (~4,600 km$^2$) of the total glacier area. Overall, glaciers of SV are known to have been losing mass over the past decades, the most recent estimate of the climatic mass balance is -8 Gt a$^{-1}$, or ca. -0.23 m w.e. a$^{-1}$, for the period 2000-2019 (Schuler et al., 2020). The larger glaciers of SV are typically polythermal, with a temperate base and cold surface layer, whereas the small glaciers are predominantly cold, possibly with only patches of temperate bed (Hagen et al., 1993; Sevestre et al., 2015). Thus, mountain glaciers display low ice velocities, typically on the order of 1-15 m a$^{-1}$ horizontally (e.g. Nutall and Hodgkins, 2005; Hambrey et al., 2005; Małecki, 2014; Lamsters et al., 2022). Such slow-flowing glaciers exhibit also low emergence/submergence velocities, up to ca. 0.2 m a$^{-1}$, indicating that their elevation changes result primarily from surface processes, rather than ice dynamics (Melvold and Hagen, 1998; Hagen et al., 2005). However, glacier surges are common in SV (e.g. Jiskoot et al., 2000; Sevestre et al., 2015), but these have been seldomly reported for local mountain glaciers over the past several decades.

NZ comprises two main mountainous parts, Yuzhny Island and Severny Island. On the southern Yuzhny Island, only mountain glaciers are found, whereas the northern Severny Island is occupied by a large ice cap drained by outlet glaciers entering valleys or fjords, with a few independent mountain glaciers and an overall ice area of 22,000 km$^2$. Mountain glaciers comprise 82 % of the population and occupy 7 % (~1,600 km$^2$) of the glacier area. Similarly, as in the case of SV, glacier surging occurs also in NZ (Grant et al., 2009), but is rare for small ice masses. FJ is a group of smaller islands covered nearly exclusively by marine-terminating ice caps, so small land-terminating glaciers (or ice caps' sectors) are sporadic (24 % of the total number and 4 %, or ~550 km$^2$, of ice area). The total ice area in FJ is ca. 13,000 km$^2$. Based on the climatic conditions it may be assumed that the small ice masses of NZ and FJ are cold-based and, therefore, slow-flowing, comparably to those in SV. Previous remote sensing investigations revealed NZ and FJ have been losing ice over the past decades. This mass loss accelerated post-2010 to -14 Gt a$^{-1}$ (or -0.68 m w.e.) and -4 Gt a$^{-1}$ (or -0.35 m w.e.), respectively for NZ and FJ (Ciracì et al., 2018; Zheng et al., 2018). Studies by

Hugonnet et al. (2021) and Sommer et al. (2022) confirmed that mountain glaciers have been thinning with spatially variable rates in both of these regions.

## 3 Methods

### 3.1 Glacier selection, outlines and regionalisation

Within each of the study regions, additional subregions were distinguished (five, two and two, respectively for SV, NZ and FJ), all comprising between two and five sites. Overall, mountain glaciers at 29 sites were analysed: 19 sites in SV, 6 in NZ and 4 in FJ (Fig. 4). In this study 'mountain glaciers' refer to small (from ca. 0.5 to ca. 30 km$^2$) land-terminating ice masses, which were selected for the analysis based on the glacier attributes provided in the RGI v6.0 (Pfeffer et al., 2014). The 382 selected glaciers, covering in total 1,373 km$^2$ or ca. 20 % of the total mountain glacier area in the Barents Sea sector, represent the dominant type

of small ice masses at a given site, e.g. niche, cirque and valley glaciers in western and central SV and in NZ, but possibly also small outlet glaciers or lobes of small ice caps in eastern SV and FJ. The choice of individual study sites is a compromise between scientific interest and data availability and aimed to cover all regions with a relatively even array of data and to encompass a wide range of glacier settings and glacier change patterns.

Glacier outlines were manually digitized from optical Sentinel-2 satellite imagery at 5 m resolution taken at or close to the date

of the later digital elevation model used to assess glacier elevation changes. Therefore, the glacier elevation change analysis does not consider areas deglaciated over the study period, which was preferred to measure realistic glacier elevation change rates close to their margins. Ice divides were typically taken from RGI v6.0 (Pfeffer et al., 2014) unless contour lines generated from recent high-resolution data suggested considerable differences. In such cases, ice divides were manually mapped. For the purpose of calculation of various glacier change parameters, areas of individual elevation bins were measured with a prescribed

uncertainty of 10 % and summed up to obtain the total glacier area at each site.

### 3.2 Digital elevation models

The basis for the glacier elevation change analysis is the high-resolution (2 m) digital elevation model (DEM) strip dataset developed by the Polar Geospatial Center (University of Minnesota) and downloaded from the ArcticDEM v3.0 repository

(https://www.pgc.umn.edu/data/arcticdem). These DEMs were automatically generated from optical satellite images at a resolution of ~0.5 m (WorldView), supported by ground control points obtained from ICE-Sat2 laser altimetry data, and provide elevations above the ellipsoid (Porter et al., 2018; Noh and Howat, 2015). ArcticDEM data are a useful tool in glacier monitoring and have been used previously in various glaciological applications (e.g. Barr et al. 2018; Zheng et al., 2018; Błaszczyk et al., 2019; Szafraniec and Dobiński, 2020; Holmlund, 2020; Elagina et al., 2021). However, the data suffer from several problems,

making their use in glacier elevation change analysis challenging over some areas. Besides the usual issues common for glacier DEM comparisons (e.g. misalignments in $x$, $y$ and $z$ axes, reduced elevation data quality over snow-covered areas due to low

contrast on satellite images etc.), ArcticDEMs contain numerous artefacts (i.e. erroneously-generated random landforms), data gaps and incomplete temporal coverage (e.g. local shortages of quality DEMs for summer seasons).

All DEMs used in this study were constructed from summer imagery (June-September) to minimize the impact of winter snow cover evolution on actual glacier surface elevation changes. The majority of source stereo-imagery was acquired between 2011 and 2017 and individual DEM pairs cover periods spanning from 3 to 7 years. At two locations in central SV (Sites 12 and 13), however, the Norwegian Polar Institute (NPI) 5 m DEMs valid for 2009 (NPI, 2014) were used as a reference due to the scarcity of summer ArcticDEMs at these locations. This introduced a mismatch between the ellipsoid height of ArcticDEM and the orthometric height of NPI DEMs. To remedy this, the ArcticDEM data were converted to orthometric elevations using the EGM2008 geoid model (Pavlis et al., 2012), which fits well the difference between the ellipsoid and orthometric heights at these concrete locations. A list of all DEMs used in this study can be found in the supplementary data (Tab. S1).

### 3.3 DEM differencing

The majority of the 2-metre DEM pairs were misaligned against each other on the order of several metres in all dimensions. To align the DEMs, the co-registration technique by Nuth and Kääb (2011) was used; however, this procedure was improved manually, since numerous data gaps and random artefacts reduced the usefulness of automatic co-registration. Alignment of DEMs was followed by subtraction of the older DEM from the more recent one to measure elevation change or elevation difference ($dh$) over the study period. The resulting rasters of $dh$ (DEMs of Differences, DoDs) were further inspected for biases with the use of test polygons located on stable, gentle surfaces with no detected artefacts (e.g. roches moutonnées, inactive outwash fans, gentle slopes, near-coast lowlands etc.). Each DEM pair was covered with 4 - 10 test polygons, all having equal area (ranging from ~80,000 to ~2,000,000 m$^2$, depending on the site) and distributed across entire areas of interest. Within the polygons, average $dh$ was calculated and added/subtracted from DoDs, depending on the sign of bias. The standard deviation of elevation differences within test polygons was further used for uncertainty assessment of $dh$ measurement (see Sect. 3.8).

### 3.4 Artefacts removal

Artefacts on glacier surfaces were manually removed by inspecting DEM hillshades for suspicious landforms (e.g. steep bumps, hollows, ridges or gullies on otherwise gentle glacier surfaces), and DoDs for unlikely patterns of glacier surface change (e.g. sharply defined areas of abnormally high or low $dh$). In some areas, the resulting data gaps were larger than areas with useful data, leaving some glacier DoDs with limited spatial coverage, down to 23 %. For more information on the impact of the spatial coverage on glacier mass balance calculations see Sect. 3.8.

### 3.5 Quantification of glacier geodetic balance

Due to the fragmentation of DoDs, the geodetic balance of glaciers at each site ($\overline{dh}/dt$) was quantified by collecting all available $dh$ data within individual 50 m elevation bins and their integration with overall glacier hypsometry, i.e. the area-altitude

distribution. To obtain elevations of each DoDs' cell, 10-metre resolution ArcticDEM mosaics were used. The mosaics are nearly complete DEMs of the Arctic constructed from the optimal ArcticDEM strips available for each area, and the surface at an unknown time, but most likely at some point between 2011 and 2017. The mosaics were downsampled to 20 m with bilinear interpolation and corrected to orthometric heights using the EGM2008 geoid model (Pavlis et al., 2012), which is accurate to within a few meters over the study regions, and is therefore sufficient for plotting the calculated glacier elevation changes against altitude.

To improve computation efficiency, de-biased and de-artefacted DoDs at 2-metre resolution were downsampled and fit to a 20 m grid of the downsampled ArcticDEM mosaics, using bilinear interpolation. All glacier grid points with *dh* data were exported with their respective orthometric elevations and aggregated into 50 m elevation bins. Statistics of elevation changes were calculated for all bins and comprise medians, arithmetic means, standard deviations, 5th, 10th, 25th, 75th, 90th and 95th percentiles, all of which can be found in the supplementary data (Tabs. S2-S30).

The hypsometry for each glacier was calculated from the glacier outlines and the downsampled and orthometrically corrected ArcticDEM mosaic data. Median values of *dh* were calculated for each 50-m elevation bin, since the median is a more robust metric than the average, when there are outliers in the data, e.g. caused by previously undetected artefacts, particularly at low sample sizes. The medians were subsequently multiplied by bin areas to obtain zonal volume change. Finally, overall glacier volume change within each site was divided by the total area of local glaciers to obtain site-wide elevation change and further recalculated to obtain annual rates, thus obtaining $\overline{dh}/dt$.

## 3.6 Conversion of elevation changes to mass balance

Conversion of $\overline{dh}/dt$ to mass balance (*B*, given in metres of water equivalent, m w.e.) is typically performed by applying a fixed density to the overall $\overline{dh}/dt$, since this approach cancels the dynamic component of elevation changes of glaciers, i.e. ice emergence or submergence. However, Huss (2013) advised caution when using such a conversion method for studies of glacier change over periods of just several years and when glaciers are close to balance. For this reason, a different attempt was used in this study.

Because glaciers studied in this paper are small and are expected to display very low horizontal and vertical ice velocities, i.e. metres per year and centimetres per year, respectively, local thinning or thickening is assumed to roughly reflect point mass balances. For this reason, the conversion was performed for individual elevation bins at each site, rather than for overall glacier areas, and further integrated over the whole hypsometry to obtain *B*. It was assumed that bins dominated by surface lowering are losing ice at a density of $890 \pm 20$ kg m$^{-3}$, whereas those dominated by elevation gains are considered as firn build-up at $550 \pm 50$ kg m$^{-3}$. This implies that the procedure does not account for firn compaction, a process that might contribute to a slight surface lowering in high-elevation areas of some glaciers.

## 3.7 Approximation of glacier volume change rates

To approximate ice volume within individual sites, the empirical scaling of Martín-Español et al. (2015) was used, which is based on radar soundings of glaciers in SV. Volumes ($V$, in km$^3$) of all glaciers used in this study were calculated using their areas ($A$, in km$^2$) and the 'logmse' scaling law:

$$V = 0.0343 \cdot A^{1.329},$$

(1)

Subsequently, all glacier volumes were summed to obtain the total ice volume for each site. Errors of volumes obtained by scaling might be large for individual glaciers, as well as for samples of hundreds (Farinotti and Huss, 2013), so a conservative uncertainty of the total site-specific $V$ approximation is set at $\pm 50$ %. The overall $V$ at the sites was then used to calculate relative volume change rates ($\overline{dv/dt}$) based on $\overline{dh/dt}$.

## 3.8 Uncertainties

Uncertainties of $dh$ calculated by DEM differencing are largely related to the quality of DEMs and the sample size used, but in this study, there is another important factor. Overall, 804 km$^2$ of glacier ice was investigated, equalling 59 % of the total area of the study glaciers (1,373 km$^2$). No DoD used has complete data coverage for the local population of glaciers, which ranges from 23 % to 98 % of their area. This implies individual glacier sites required different magnitudes of data extrapolation from surveyed to unsurveyed areas.

Considering above, it was assumed that uncertainties of elevation changes ($\varepsilon$) measured for individual elevation bins are a function of: (i) the standard deviations of elevation differences within test polygons on low-relief stable ground ($\sigma$), representing the uncertainty of $dh$ measurement of a single cell; (ii) the total area surveyed within each elevation bin in km$^2$ representing the sample size ($N$) after Nuth et al. (2007); and (iii) the percentage of the unsurveyed area of each bin ($A_u$), so that bins with low coverage have higher uncertainty.

$$\varepsilon = \frac{1.5\sigma}{\sqrt{N}} \cdot (1 + A_u),$$

(2)

The multiplicator 1.5 in (2) aims to account for potentially lower quality of elevation data of ice and snow surfaces due to their lower contrast on satellite images. In cases where the lowest or highest bins of the local glacier population had insufficient data to calculate reliable statistics (less than 100 cells), $dh$ data from the nearest useful bin were used instead, but with doubled uncertainty. Alternatively, missing elevation bins located in the middle of glaciers were approximated for $dh$ by averaging statistics from two neighbouring bins. Having the uncertainties of $dh$ for each elevation bin, uncertainties for other derived glacier change parameters were calculated by conventional error propagation techniques.

Overall, the estimated $\overline{dh/dt}$ uncertainties for individual sites range from $\pm 0.04$ to $\pm 0.19$ m a$^{-1}$. However, these values might be much higher for some elevation bins (compare with Tabs. S2-S30). This commonly results from the small size of elevation

bins, further boosted to metre-scale uncertainty values at low data coverage. However, large uncertainties typically refer only to a few bins within each site, with a very low proportion in the overall glacier area. For this reason, these do not have a large impact on the overall $\overline{dh}/dt$ uncertainty.

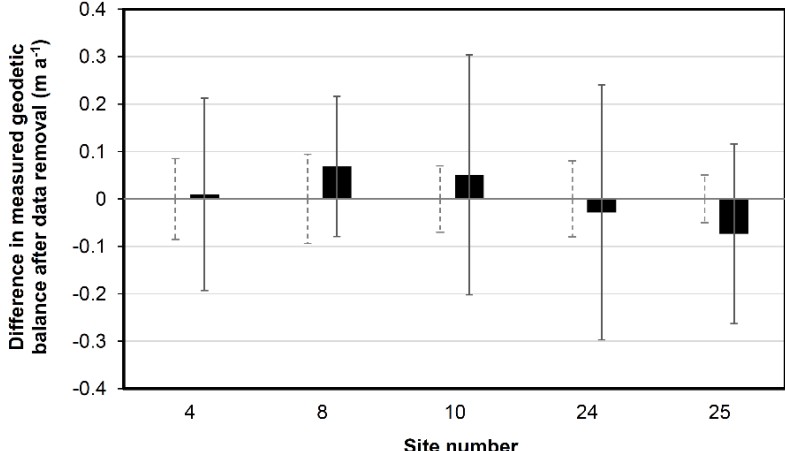

**Figure 3 Results of the sensitivity test performed on sites with data coverage > 90 %. Black bars – differences in the calculated glacier geodetic balances between the full-data coverage and reduced-data coverage; grey whiskers – uncertainty range of the reduced-data coverage balances; dashed whiskers – uncertainty range of the full-data coverage balances. For data coverage used in the test see Fig. S1.**

To quantify the possible distortions of $\overline{dh}/dt$ calculations related to the low data availability, a sensitivity test was performed on five sites with the best data coverage in SV and NZ (Sites 4, 8, 10, 24 and 25). Much data from respective glacier DoDs were manually removed, so that the data coverage dropped from the original 92 – 95 % down to ca. 20 – 25 % (Fig. S1). Subsequently, $\overline{dh}/dt$ was recalculated for individual sites, together with its uncertainties. The balances calculated with fewer data changed only slightly, within the range from -0.07 to 0.07 m a⁻¹ and with an arithmetic mean of 0.01 m a⁻¹, and typically stayed within the original uncertainty range (Fig. 3). Unsurprisingly, data removal had a larger effect on the recalculated uncertainties which increased by a factor of 2 to 4 times.

## 4 Results

The results of DEM differencing are summarized in Tab. 1 and Figs. 4, 5 and 6. For detailed information for each site see the supplementary data, i.e. statistics (Tabs. S2-S30) and figures (relationships between elevation change and altitude in Figs. S2-S30 and maps in Figs. S31-S43).

### 4.1 Svalbard

SV contains the largest population of mountain glaciers across the European Arctic. The most negative $\overline{dh}/dt$ was found in SV-W (Fig. 4a), where all glaciers have been experiencing thinning, and at all elevations. Relationships between glacier elevation

change and altitude were fairly coherent across this subregion (Fig. 5a) so that the site-to-site variability in $\overline{dh/dt}$ (from -1.58 m a$^{-1}$ at Site 3 to -0.87 m a$^{-1}$ at Site 5) resulted largely from hypsometry, here represented by the median elevation (Tab. 1; Fig. 6).

**Table 1 General statistics of glacier geometry, elevation change, mass balance and volume change across the study sites (for site locations see Fig. 4).**

| Region and subregion | Site number | Site name | Total area of the study glaciers | Glacier area surveyed | Median elevation of the study glaciers | Date of first DEM | Date of last DEM | Standard deviation of DEM elevation differences on test polygons, $\sigma$ | Geodetic balance, $\overline{dh/dt}$ | Total area of elevation bins with positive change | Mass balance, $B$ | Relative volume change rate, $\overline{dv/dt}$ | Approximate time of complete decay assuming constant relative volume change rate |
|---|---|---|---|---|---|---|---|---|---|---|---|---|---|
| | | | [km$^2$] | [%] | [m] | | | [m] | [m a$^{-1}$] | [%] | [m w.e. a$^{-1}$] | [% a$^{-1}$] | |
| **Svalbard** | | | | | | | | | | | | | |
| SV-NW | 1 | Magdalenafjorden | 27 | 40 | 268 | 2012/07/01 | 2016/07/08 | 0.48 | -0.41 ± 0.09 | 5 | -0.37 ± 0.08 | -0.94 ± 0.52 | early-22nd century |
| SV-NW | 2 | Forlandsundet | 72 | 62 | 335 | 2013/09/15 | 2017/09/14 | 0.56 | -0.62 ± 0.06 | 5 | -0.55 ± 0.05 | -1.11 ± 0.56 | early-22nd century |
| SV-W | 3 | Prins Karls Forland | 15 | 61 | 174 | 2011/07/15 | 2017/08/02 | 1.07 | -1.58 ± 0.16 | 0 | -1.40 ± 0.14 | -3.91 ± 1.99 | 2040s |
| SV-W | 4 | Grønfjorden | 53 | 92 | 295 | 2013/08/17 | 2016/07/25 | 0.75 | -1.17 ± 0.09 | 0 | -1.05 ± 0.08 | -1.85 ± 0.93 | 2070s |
| SV-W | 5 | Van Keulenfjorden | 78 | 57 | 445 | 2013/07/30 | 2016/07/26 | 0.56 | -0.87 ± 0.13 | 0 | -0.77 ± 0.11 | -0.93 ± 0.48 | early-22nd century |
| SV-W | 6 | Nottinghambukta | 37 | 62 | 360 | 2013/07/30 | 2016/07/26 | 0.75 | -1.05 ± 0.19 | 0 | -0.93 ± 0.17 | -1.18 ± 0.62 | early-22nd century |
| SV-W | 7 | Sørkapp | 45 | 87 | 248 | 2013/08/11 | 2017/08/03 | 0.39 | -1.49 ± 0.08 | 0 | -1.33 ± 0.07 | -1.95 ± 0.98 | 2060s |
| SV-E | 8 | Edgeøya South | 39 | 95 | 278 | 2012/08/08 | 2017/07/21 | 0.59 | -1.56 ± 0.09 | 0 | -1.39 ± 0.08 | -3.01 ± 1.51 | 2050s |
| SV-E | 9 | Edgeøya North | 86 | 68 | 348 | 2012/06/02 | 2017/06/25 | 0.67 | -0.85 ± 0.06 | 0 | -0.76 ± 0.06 | -1.41 ± 0.71 | 2080s |
| SV-E | 10 | Bjørnsundet | 12 | 94 | 245 | 2011/08/18 | 2017/07/25 | 0.74 | -0.58 ± 0.07 | 0 | -0.51 ± 0.06 | -1.05 ± 0.54 | early-22nd century |
| SV-C | 11 | Agardhbukta | 103 | 39 | 408 | 2013/07/14 | 2017/06/30 | 0.55 | -0.72 ± 0.07 | 0 | -0.64 ± 0.07 | -0.99 ± 0.50 | early-22nd century |
| SV-C | 12 | Nordenskiöld Land | 51 | 57 | 591 | Early-July 2009 | 2013/07/17 | 0.65 | -0.37 ± 0.07 | 0 | -0.33 ± 0.07 | -0.64 ± 0.34 | late-22nd century |
| SV-C | 13 | Dickson Land | 48 | 74 | 497 | Early-July 2009 | 2013/07/17 | 0.93 | -0.57 ± 0.08 | ~0 | -0.50 ± 0.07 | -0.95 ± 0.49 | early-22nd century |
| SV-C | 14 | Atomfjella | 107 | 43 | 784 | 2013/09/01 | 2017/09/10 | 0.61 | -0.17 ± 0.07 | 43 | -0.16 ± 0.04 | -0.19 ± 0.12 | post-22nd century |
| SV-C | 15 | Vestfjorden | 90 | 23 | 769 | 2011/06/15 | 2016/07/02 | 0.71 | -0.10 ± 0.11 | 43 | -0.11 ± 0.07 | -0.16 ± 0.18 | post-22nd century |
| SV-N | 16 | Primatesfjella | 61 | 62 | 403 | 2012/08/15 | 2017/07/10 | 0.77 | 0.26 ± 0.06 | 85 | 0.14 ± 0.03 | 0.46 ± 0.25 | n/a |
| SV-N | 17 | Andrée Land | 112 | 57 | 480 | 2012/07/07 | 2017/07/31 | 0.65 | 0.08 ± 0.04 | 81 | 0.03 ± 0.02 | 0.14 ± 0.10 | n/a |
| SV-N | 18 | Laponiahalvøya | 14 | 43 | 229 | 2012/06/15 | 2017/08/06 | 0.58 | -0.18 ± 0.12 | 39 | -0.17 ± 0.09 | -0.41 ± 0.35 | post-22nd century |
| SV-N | 19 | Platenhalvøya | 11 | 72 | 172 | 2013/07/13 | 2017/07/23 | 0.68 | -0.14 ± 0.15 | 38 | -0.14 ± 0.10 | -0.23 ± 0.27 | post-22nd century |
| **Novaya Zemlya** | | | | | | | | | | | | | |
| NZ-S | 20 | Yuzhny Island | 53 | 53 | 569 | 2013/06/18 | 2016/06/17 | 0.44 | -1.23 ± 0.09 | 0 | -1.09 ± 0.08 | -1.76 ± 0.89 | 2070's |
| NZ-S | 21 | Rusanov Valley | 48 | 40 | 471 | 2009/07/26 | 2016/08/17 | 0.65 | -1.27 ± 0.09 | 0 | -1.13 ± 0.08 | -2.17 ± 1.10 | 2060's |
| NZ-S | 22 | Mashigin Fjord | 60 | 39 | 512 | 2012/07/20 | 2015/09/28 | 0.52 | -1.28 ± 0.12 | 0 | -1.14 ± 0.11 | -2.16 ± 1.10 | 2060's |
| NZ-N | 23 | Nordenskiöld Bay | 52 | 67 | 447 | 2012/08/29 | 2016/08/09 | 0.39 | -0.87 ± 0.05 | 0 | -0.78 ± 0.05 | -1.20 ± 0.60 | early-22nd century |
| NZ-N | 24 | Borzov Bay | 12 | 94 | 418 | 2013/07/31 | 2017/09/08 | 0.59 | -0.81 ± 0.08 | 0 | -0.72 ± 0.07 | -1.61 ± 0.82 | 2070's |
| NZ-N | 25 | Inostrantsev Bay | 11 | 93 | 467 | 2013/08/03 | 2017/07/20 | 0.35 | -0.49 ± 0.05 | 0 | -0.43 ± 0.04 | -0.80 ± 0.41 | mid-22nd century |
| **Franz Josef Land** | | | | | | | | | | | | | |
| FJ-SW | 26 | Prince George Island | 21 | 98 | 145 | 2012/08/13 | 2017/08/13 | 0.44 | -0.34 ± 0.04 | 0 | -0.30 ± 0.03 | -0.44 ± 0.22 | post-22nd century |
| FJ-SW | 27 | Markham Sound | 27 | 84 | 152 | 2013/07/19 | 2017/07/18 | 0.37 | -0.47 ± 0.04 | 0 | -0.42 ± 0.04 | -1.10 ± 0.56 | early-22nd century |
| FJ-NE | 28 | Wilczek Land | 10 | 76 | 299 | 2013/07/19 | 2017/07/18 | 0.62 | -0.17 ± 0.11 | 8 | -0.15 ± 0.09 | -0.29 ± 0.24 | post-22nd century |
| FJ-NE | 29 | Jackson Island | 20 | 73 | 118 | 2013/08/29 | 2017/08/13 | 0.76 | -0.03 ± 0.11 | 55 | -0.04 ± 0.09 | -0.05 ± 0.19 | post-22nd century |

In SV-E thinning was also glacier-wide, supported by the relatively low elevations of the mountain glaciers. There is a northward trend of less negative elevations change from the available data. Glacier $\overline{dh/dt}$ ranged from -1.56 m a$^{-1}$ in the south (Site 8) to -0.58 m a$^{-1}$ in the north (Site 10).

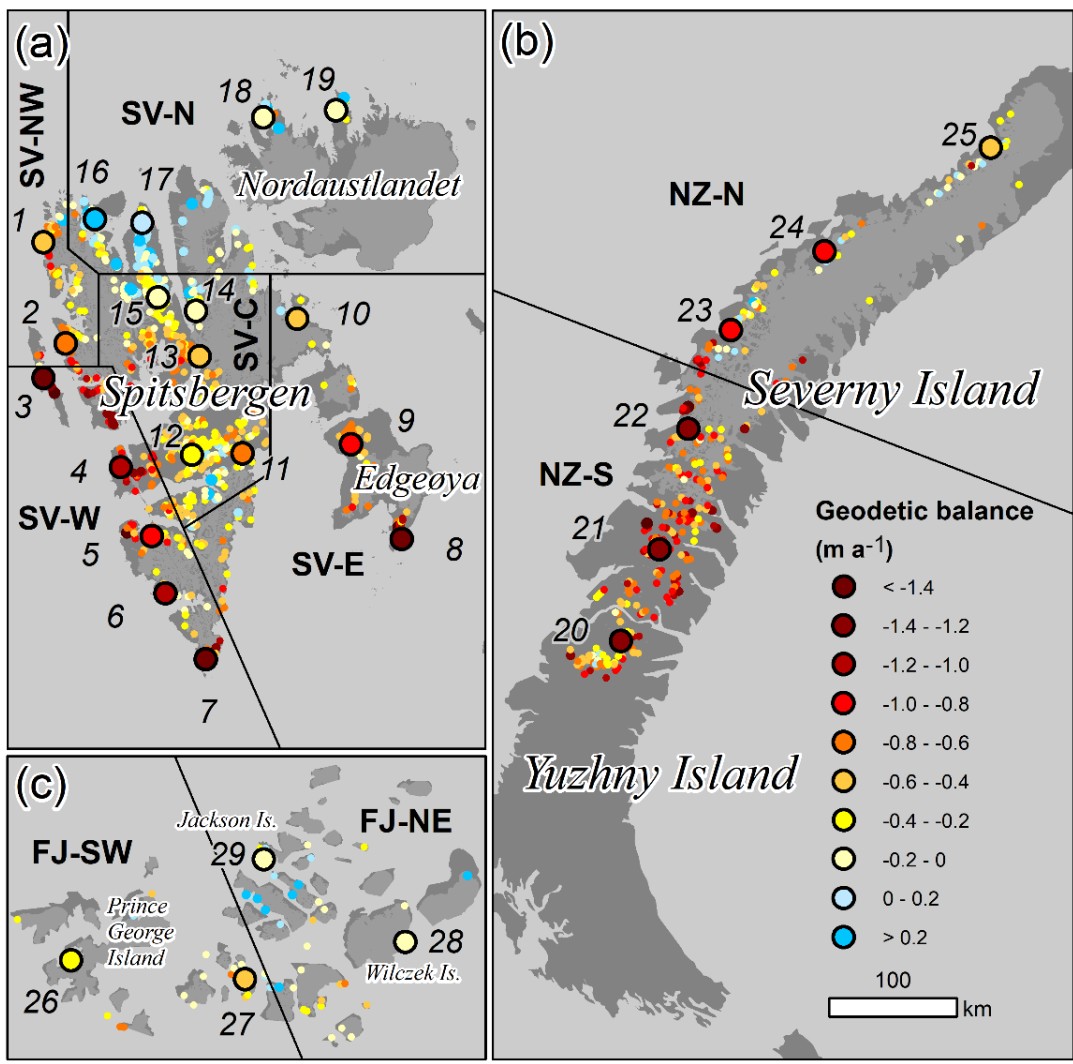

**Figure 4 Geodetic balance of mountain glaciers for the study sites (large circles and numbers) on (a) Svalbard, (b) Novaya Zemlya and (c) Franz Josef Land. Dots indicate individual mountain glaciers with colours corresponding to their geodetic balance, based on the the Randolph Glacier Inventory (Pfeffer et al., 2014) and the global glacier elevation change study by Hugonnet et al. (2021), to provide background information. For site-specific details and maps see Tab. 1 and the supplementary data.**

The subregion containing the highest elevation ice masses presented in this study is SV-C. These showed positive *dh* only above ca. 900 m a.s.l. (Fig. 5a), so many glaciers not exceeding this altitude have been thinning in all elevation bins. Despite the highest area-altitude distribution, mountain glaciers of SV-C have been generally losing mass, with $\overline{dh/dt}$ spanning from -0.72 m a$^{-1}$ (Site 11) to -0.10 m a$^{-1}$ (Site 15), strongly correlated with glacier median elevations (Tab. 1; Fig. 6).

In SV-N, mountain glaciers have a contrasting pattern of change; here, glaciers have had positive *dh* above 225-360 m a.s.l. (Fig. 5a). Overall, glaciers in SV-N were close to geodetic equilibrium or even thickened. Glacier $\overline{dh}/dt$ at four sites ranged from -0.18 to -0.14 m a$^{-1}$ in north Nordaustlandet (Sites 18 and 19, respectively) to 0.08 to 0.26 m a$^{-1}$ in north Spitsbergen (Sites 17 and 16).

The subregion SV-NW might be considered a transition zone between rapid ice loss of SV-W and balanced conditions of SV-N,
with a poleward gradient reducing glacier thinning rates. Both sites in SV-NW showed a thickening trend above ca. 480-600 m a.s.l. (Fig. 5a), but otherwise negative $\overline{dh}/dt$ of -0.62 (Site 2) and -0.41 m a$^{-1}$ (Site 1).

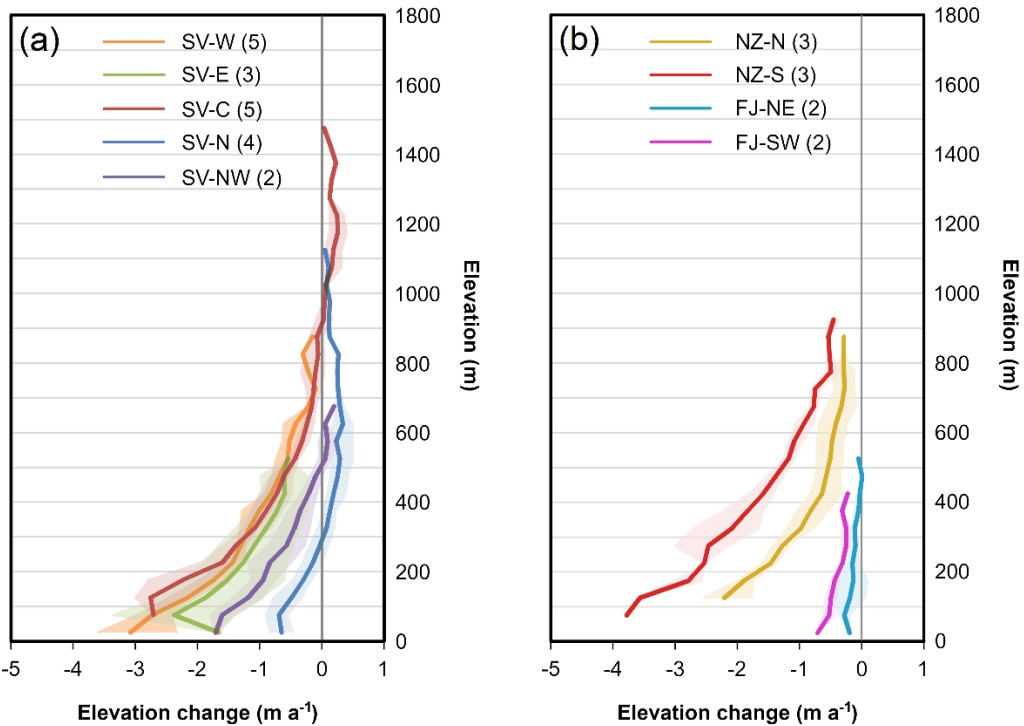

**Figure 5 Relationships between glacier elevation change and altitude averaged over subregions for (a) Svalbard and (b) Novaya Zemlya and Franz Josef Land. Numbers in parentheses denote the number of sites within each subregion, while shading indicates minimum**
**and maximum elevation change values at a given altitude within each subregion. For site-specific glacier elevation change curves and glacier hypsometry, see the supplementary data.**

## 4.2 Novaya Zemlya

In the subregion NZ-N, all three sites of mountain glaciers showed glacier-wide thinning, with an apparent northward trend of less negative $\overline{dh}/dt$, ranging from -0.87 m a$^{-1}$ at Site 23 to -0.49 m a$^{-1}$ at Site 25. Three sites, 20, 21 and 22, comprising the other
subregion, NZ-S, had higher rates of glacier-wide mass loss, with $\overline{dh}/dt$ between -1.28 m a$^{-1}$ (Site 22) and -1.23 m a$^{-1}$ (Site 20)(Fig. 4b), and rapid marginal thinning (Fig. 5b).

**4.3 Franz Josef Land**

The FJ region is characterized by low topography and extensive large ice caps, on most islands. For these reasons, the region contains the fewest mountain glaciers, so to collect sufficient data, some ice masses analysed here were small land-terminating

ice caps or their gentle lobes, rather than cirque or valley glaciers. In general, two sites of small glaciers in FJ-NE have been relatively close to balance in most elevation bins, with total $\overline{dh}/dt$ of -0.17 m a$^{-1}$ (Site 28) and -0.03 m a$^{-1}$ (Site 29). Moreover, just south of Site 29 the global data by Hugonnet et al. (2021) suggests a small area with positive changes (Fig. 4c). On the other hand, moderate but glacier-wide thickness loss was observed at two sites in FJ-SW, with $\overline{dh}/dt$ of -0.47 m a$^{-1}$ at Site 26 and - 0.34 m a$^{-1}$ at Site 27 (Figs. 4c and 5b)

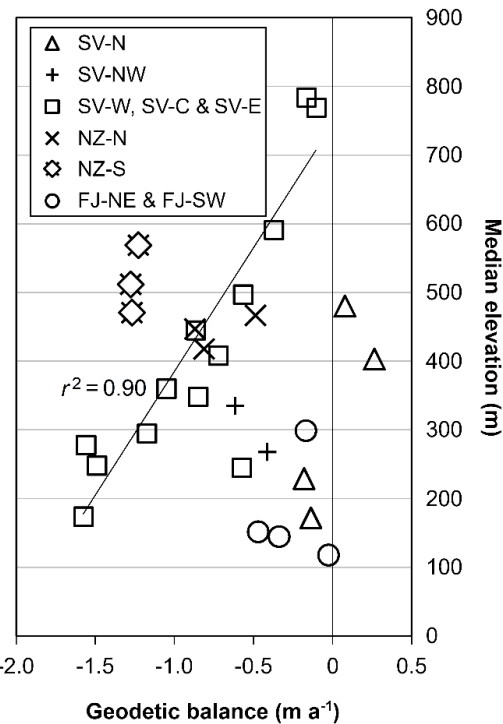

**Figure 6 Relationships between site-wide geodetic balances of glaciers and their median elevations. The trend line and its squared correlation correspond to SV-W, SV-C and SV-E, excluding a single outlier (Site 10).**

**5 Discussion**

**5.1 Spatial variability of glacier change**

Over the study period, mountain glaciers across the European Arctic have been generally losing mass. On a regional scale, the fastest rates of mass loss were observed in NZ, while losses were modest in FJ. The median $B$ calculated from data in Tab. 1 was -0.51 m w.e. for sites in SV, -0.93 m w.e. for NZ and -0.23 m w.e. for FJ. Application of these median values to the ~4600 km$^2$

covered by mountain glaciers in SV, ~1600 km$^2$ in NZ and ~550 km$^2$ in FJ (RGI v6.0, Pfeffer et al., 2014) yields rough estimates of their relative contributions to total mass balances of individual regions; ca. 25 %, 10 % and 2 %, respectively, if taking the recently published totals of -8 Gt a$^{-1}$ (or -0.23 m w.e.) for SV, -14 Gt a$^{-1}$ (or -0.64 m w.e.) for NZ and -4 Gt a$^{-1}$ (or -0.35 m w.e.) for FJ (Schuler et al., 2020; Ciracì et al., 2018; Zheng et al., 2018).

Comparison of $B$ reported in Tab. 1 against the regional totals summarised above highlights the generally faster melting of the small ice masses. Ice losses were particularly strong along western Spitsbergen (SV-W), Edgeøya (part of SV-E) and NZ (NZ-S and NZ-N). This partly correlates with the general trajectory of Atlantic currents (Fig. 1a), recent strong warming of the sea surface (Morris et al., 2020) and, to some extent, with a sharp reduction of winter sea ice concentration (Fig. 2c). This suggests that the aforementioned processes of Atlantification contribute to changes of land-terminating glaciers on islands surrounding the Barents Sea, but this would require further testing.

The greatest spatial variability of $\overline{dh/dt}$ was found in SV, with two trends in thinning rates (Fig. 4a). There is an overall trend of less negative/more positive $\overline{dh/dt}$ to the north, and another to the interior of SV-C. However, the relationships between glacier thinning and altitude were remarkably consistent between SV-W, SV-C and partly SV-E (Fig. 5), so at a common elevation their thinning rates have been similar (Fig. S43), and close to zero at ca. 900 m a.s.l. This implies that over much of SV (SV-W, SV-C and SV-E), it is predominantly median elevation that controls spatial variability of $\overline{dh/dt}$ of mountain glaciers (with linear regression at $r^2 = 0.90$ and slope of 0.25 m a$^{-1}$ 100 m$^{-1}$)(Fig. 6), rather than climatic differences, e.g. associated with the distance to the open sea. In NZ a clear northward gradient of less negative $\overline{dh/dt}$ was also found, whereas in FJ $\overline{dh/dt}$ was more homogenous, although also with a possible slight trend of reducing thinning rates to the northeast (Figs. 4b and 4c).

**5.2 Glacier-wide thinning and relative volume changes**

As many as 21 of the 29 sites investigated have been experiencing substantial thinning ($\overline{dh/dt} < $ -0.20 m a$^{-1}$). At 19 of these sites, $dh$ was negative, even at the highest elevations (Tab. 1). This glacier-wide thinning is apparent in all study regions and as such might be regarded as the dominant style of the response of mountain glaciers to recent climatic conditions across the Barents Sea area.

Considering the low dynamic activity of the study glaciers and the dominant role of surface processes to their elevation changes, the glacier-wide thinning suggests that many of these glaciers would eventually melt away completely even without further warming. Recent $\overline{dv/dt}$ across all regions was on average close to -1 % a$^{-1}$, but some subregions have been experiencing far more negative $\overline{dv/dt}$, particularly SV-W, SV-E and NZ-S, where some sites had losses of 2-4 % a$^{-1}$ (Tab. 1).

Assuming a continuation of the observed $\overline{dv/dt}$ into the future, mountain glaciers might disappear from many study sites within the coming two to five decades, whereas glaciers in the majority of the sites might vanish within about 100 years (Tab. 1). Although the glacier changes were inferred for a relatively short period and the assumption of constant $\overline{dv/dt}$ oversimplifies the processes behind glacier and climate evolution, the numbers provided above are still indicative of the critical state of mountain glaciers at many sites in SV and NZ. The ultimate disappearance of mountain glaciers will shift individual valleys or

entire subregions from glacierized to ice-free, e.g. large parts of SV-C or NZ-S. This might have a large impact on the landscape, fjord systems, land hydrology and ecology, among others (e.g. Milner et al., 2017; Huss and Hock, 2018; Strzelecki et al., 2018; 2020; Cauvy-Fraunié and Dangles, 2019; Torsvik et al., 2019; Hopwood et al., 2020).

## 5.3 Comparison with other studies

The glacier change trends described in this paper are in general agreement with previously published case studies for several
areas in SV, which indicate mass loss from smaller glaciers and high-elevation thinning, particularly in SV-W, SV-NW and SV-C (e.g. Kohler et al., 2007; James et al., 2012; Małecki, 2016; Sobota et al., 2016; Elagina et al., 2021). Negative trends of change of small glaciers can be seen also in NZ and FJ, in line with Ciracì et al. (2018), Zheng et al. (2018) and Sommer et al. (2022). The global glacier $\overline{dh/dt}$ dataset at 100 m resolution by Hugonnet et al. (2021) matches well the spatial glacier change variability reported here for all three regions SV, NZ and FJ (Fig. 4). However, within the glacier boundaries used in this study, the changes
calculated with the use of the dataset by Hugonnet et al. (2021) for the period 2010-2019 were on average more positive by ca. 0.4 m a$^{-1}$ for sites experiencing strong imbalances ($\overline{dh/dt} < $ -1.0 m a$^{-1}$) and by ca. 0.1 m a$^{-1}$ more negative for sites displaying balanced conditions (sites with $\overline{dh/dt} > $ -0.2 m a$^{-1}$). This discrepancy might be linked to several factors, e.g. differences in periods of observation, resolutions, data coverage, input data, etc., the relative contribution of which is difficult to quantify. Nevertheless, the match between $\overline{dh/dt}$ reported in this study and those by Hugonnet et al. (2021) is high, even accounting for
the slightly different periods of observation and a certain level of noise in the global inventory data (Figs. 7c and S44-S53). A number of glacier mass balance modelling studies is available for SV, e.g. the recent models by van Pelt et al. (2019) and Noël et al. (2020), both at relatively high resolutions of 1000 m and 500 m, respectively, which are sufficient to resolve some of the main features of the recent behaviour of mountain glaciers. These models also reproduce $B$ values similar to $\overline{dh/dt}$ reported in this study over most locations. On several sites, however, the model by Noël et al. (2020) tends to overestimate marginal melting
(e.g. Sites 16 and 17; Figs. 7b, S50b and S51b) or high-elevation accumulation (e.g. Sites 4 and 7; Figs. S45b and S47f). The model by van Pelt et al. (2019) appears to reproduce accurate point $B$ (Figs. 7a, S44a, S45a, S47-51), although its lower resolution might overestimate the total $B$ of some mountain glacier groupings by uneven representation of high-elevation and lower glacier zones, the latter of which are commonly narrower and, therefore, ignored.

## 5.4 The North Spitsbergen anomaly

Despite the recent strong warming trend across the Barents Sea and the generally rapid ice loss among its mountain glaciers, eight sites analysed here had $\overline{dh/dt} > $ -0.20 m a$^{-1}$. All of these were restricted to the north-eastern rims of SV (SV-N and north part of SV-C) and FJ (FJ-NE). At two of these sites in north Spitsbergen (Sites 16 and 17) glaciers even gained mass over the study period (Figs. S17, S18, S38), implying a strong gradient in mass balance forcing over a short distance, i.e. from SV-NW to SV-N (Fig. 4a). This spatial pattern is in line with the data by Hugonnet et al. (2021) and the mass balance modelling by van

Pelt et al. (2019)(Figs. 7, S50 and S51), which also indicate that positive changes at Sites 16 and 17 have been a part of a more general trend in north Spitsbergen, i.e. to the east of Site 17 (Fig. 4a).

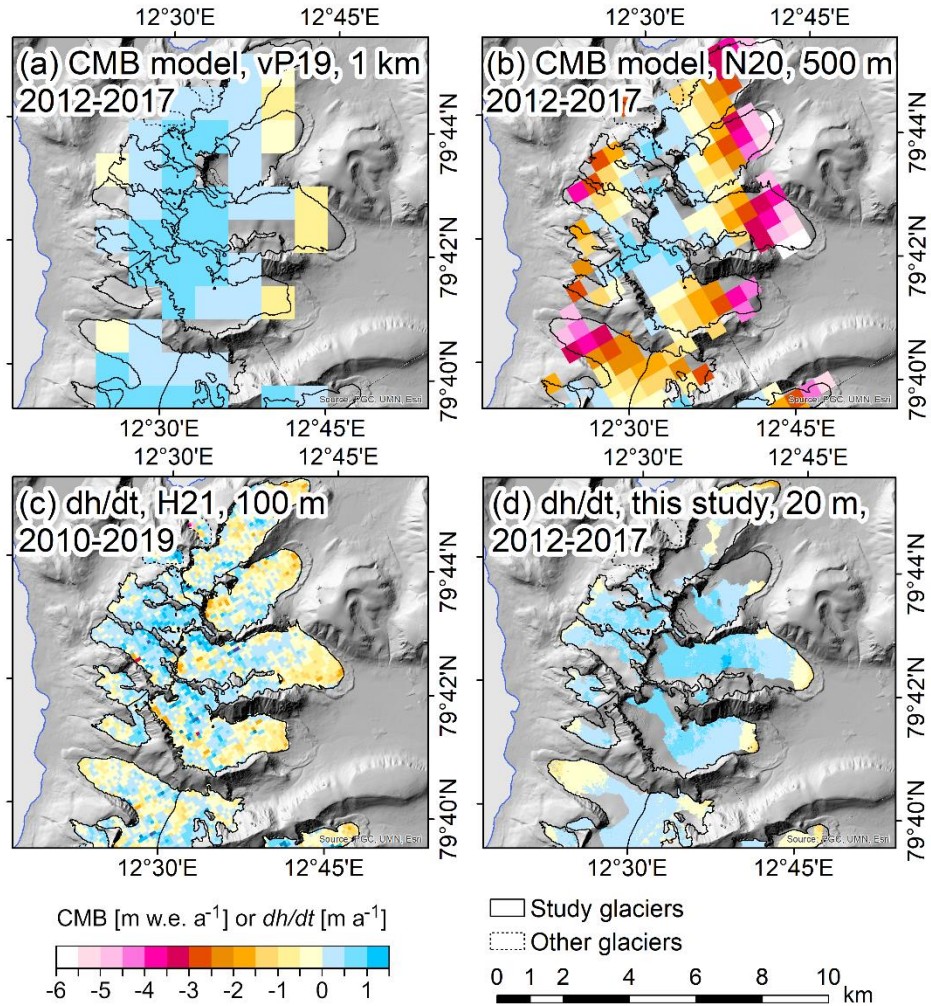

**Fig. 7 Example comparison of glacier mass balance models and elevation change observations at Site 16 in SV-N. Outcomes of climatic mass balance (CMB) models by (a) van Pelt et al. (2019) at 1000 m resolution, and (b) by Noël et al. (2020) at 500 m resolution. Observed**
$\overline{dh}/dt$ **after (c) Hugonnet et al. (2021) at 100 m resolution, and (d) this study at 20 m resolution, downsampled from 2 m. Note the colour scale is common for CMB and $\overline{dh}/dt$. Hillshaded background: ArcticDEM, source: Polar Geospatial Center, University of Minnesota, Esri.**

The duration of the north Spitsbergen anomaly remains unclear at this point, however, previously collected data suggest this
might be a relatively recent phenomenon. Positive $\overline{dh}/dt$ and *B* over the study period was accompanied by a slight retreat of glacier margins at Sites 16 and 17, while an earlier direct study from this area by Etzelmüller et al. (1993) reported a clearly

negative $B$ for one glacier at Site 16 between 1970 and the 1990s. These facts suggest that the mass gain reported in this paper resulted from a post-1990 climate forcing, acting on a shorter timescale than the reaction time of glaciers. The global study by Hugonnet et al. (2021) suggests that mountain glaciers in north Spitsbergen have been relatively close to balance at least since the early-2000's. On the other hand, in the modelled dataset at 1000 m resolution by van Pelt et al. (2019) Sites 16 and 17 show long-term positive $B$ at least since the 1950s, so contrary to the more focused study by Etzelmüller et al. (1993), but this might be partly related to the underrepresentation of narrower ablation zones.

One hypothetical explanation of the anomaly might be the recent rapid retreat of winter sea ice north of SV (Onarheim et al., 2014), potentially favouring more snowfall over SV-N. An increase in winter precipitation over 2011-2017 is fairly homogenous across SV in the coarse ERA5 reanalysis data (Fig. 2b). In contrast, finer glacier mass balance modelling by van Pelt et al. (2019) and Noël et al. (2020) show an increased snowfall at Sites 16 and 17 (by up to ca. 200 mm w.e. a$^{-1}$) over the period of this study (reference period 1981-2010), and drier conditions over some other subregions of SV. Moreover, van Pelt et al. (2019) found that SV-N has been the only subregion with a significant long-term (1957-2018) precipitation increase.

The exact drivers of the positive $\overline{dh/dt}$ at Sites 16 and 17 would require further, more detailed investigations. To date, only large ice masses (> 200 km$^2$) with extensive accumulation zones have been reported to stay close to balance in SV (e.g. Bamber et al., 2004; Schuler et al., 2020) whereas mountain glaciers have been traditionally perceived to continuously lose mass since the Little Ice Age termination in the early-20$^{th}$ century, generally at increasing rates (e.g. Kohler et al., 2007; Małecki, 2016; Sobota et al., 2016; Schuler et al., 2020). Therefore, the north Spitsbergen anomaly might be considered a surprise, regardless of the mechanisms and duration of the glacier growth. Ideally, direct monitoring of glaciers in north Spitsbergen should be considered to implement, since this area is underrepresented in the Svalbard-wide glaciological studies (Schuler et al., 2020), thus likely biasing the surface mass balance measurements towards regions with faster mass losses.

**Conclusions**

This study used high-resolution elevation data (ArcticDEM) to investigate recent (ca. 2011-2017) elevation changes of 382 small (< ca. 30 km$^2$) land-terminating glaciers (or mountain glaciers) across the European High Arctic. The inferred data might be regarded as point surface mass balance proxies, since elevation changes of Arctic mountain glaciers result mainly from surface processes, rather than ice dynamics. The study documents spatially variable patterns of change among mountain glaciers in Svalbard (SV), Novaya Zemlya (NZ) and Franz Josef Land (FJ).

The vast majority of the study glaciers has been experiencing considerable thinning, at several sites exceeding the rate of 1.0 m a$^{-1}$, particularly in SV-W, partly SV-E and NZ-S. Glacier-wide thinning has been the dominant style of response to climate warming over the Barents Sea area and strong thinning at high elevations indicate negative mass balance in their former accumulation zones. This implies that many of these small ice masses might melt away even in the present climate, regardless of future climate change mitigation efforts and climate warming rates. Under hypothetical constant volume loss rates reported

in this study, many sites in SV and NZ would nearly completely lose mountain glaciers within the coming half-century, underlining their critical imbalance with the recent climate.

In SV, having the largest population of mountain glaciers, median elevation was the primary control on glacier mass balance across most of the region, i.e. in the western, central, and eastern parts. Despite the overwhelming dominance of negative changes, many mountain glaciers at some of the northernmost sites in SV have been remaining close to equilibrium or even thickened over the period of study. The nature of this anomaly requires further investigation, particularly because the SV region has been experiencing one of the greatest climate warming rates over the past half-century globally. This finding highlights the

complexity of climate-glacier interactions, even for low-activity ice masses, and the possibility of very contrasting glacier changes over relatively short distances. The results of this research provide a useful benchmark for the calibration/validation of climate models, particularly over areas with insufficient *in situ* mass balance data.

### Data availability

The ArcticDEM digital elevation model strips and mosaics are available via the ArcticDEM repository

https://www.pgc.umn.edu/data/arcticdem/. The ERA5 reanalysis datasets are available via the Copernicus programme https://cds.climate.copernicus.eu/cdsapp#!/search?type=dataset. Detailed statistics on glacier elevation change data for mass balance model calibration/validation and related maps are available in the supplementary data. Georeferenced glacier elevation change data developed for this study (DEMs of Differences, DoDs) are available upon request from the author.

### Competing interests

The author declares no competing interests.

### Acknowledgements

I sincerely thank dr. Agata Buchwał for constructive comments on the early versions of the manuscript and the Adam Mickiewicz University and its units (the Faculty of Geographical and Geological Sciences and the Institute of Geoecology and Geoinformation) for support. Criticism of two anonymous reviewers is greatly appreciated, as it helped to improve the quality

of this paper.

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
