# Peer review of "Recent contrasting behaviour of mountain glaciers across the European High Arctic revealed by ArcticDEM data"

_The Cryosphere, 2021_

## Referee Comment (RC1)

Review of: **"Recent contrasting behaviour of mountain glaciers across the European High Arctic revealed by ArcticDEM"**, by *J. Malecki* and submitted to *The Cryosphere*.

The author uses high-resolution data from the ArcticDEM to estimate recent elevation changes (2011-2017) over a set of mountain glacier sites located in the Svalbard archipelago (SV), Novaya Zemlya (NZ) and Franz Josef Land (FJ) in the Russian Arctic. Elevation change at these sites is further used to estimate total volume and mass balance changes of all mountain glaciers covering these three regions. The author finds that most glaciers have undergone thinning since 2011. At the current mass loss rate, they could melt away by the end of the 22nd century. In contrast, two glacier sites in North Svalbard (SV-N) have recently experienced thickening, potentially linked to accumulation increase following regional sea-ice decline. The author also discusses spatial gradients in mass loss decreasing from the warmer South to the colder North, and from the low-lying margins to the elevated glacier interior.

This is a well-written and interesting study estimating elevation/mass change of small mountain glaciers in the Arctic. While these results will be of interest for the community, the reviewer has some concerns that are listed below as general, specific and stylistic comments. The reviewer deems that **major** revisions are required before publication in TC.

**General comments**

1. While estimation of uncertainties is well described in the manuscript, uncertainties in elevation/volume/mass change can be large for different glacier sites. For instance, these uncertainties in elevation/mass balance can sometimes be equal to or double the absolute change for a given elevation bin (see Supplementary Tables). In addition, the spatial coverage of surveyed glacier can be as low as ~20%, with large data gap in given elevation bins (see the hypsometry graphs in the Supplementary Figures). The authors should elaborate on how these local uncertainties impact the results estimated for the different glacier sites, as well as how low spatial coverage affects the elevation change estimates. To address this issue, a sensitivity experiment exploring the impact of low spatial coverage on the elevation change estimates could be conducted by reducing the spatial coverage of a well-surveyed site (e.g. > 90% coverage). It is important to have more insight on how much spatial coverage of glacier sites is required to provide reliable estimates.

2. The selected glacier sites only cover 1373 km$^2$, which represents ~20% of the total mountain glacier area of the three regions (L287-288), and 2% of the total ice-covered area in SV, NZ and FJ. Out of these glacier sites, 804 km$^2$ (59% of the studied area) can be used to estimate elevation/mass changes. The reviewer questions whether a 15-20% coverage of these mountain glaciers is sufficient to draw firm conclusions, notably the sharp regional contrast between site 1 and sites 16-17. For instance, are these two small glacier sites in North Svalbard representative of the surrounding northern region? The authors should elaborate on this topic.

3. Recent publications are omitted in the literature review, in particular the author claims that the spatial resolution in previous studies was not sufficient to provide reliable estimates of mass change over small mountain glaciers (L25-27). However previous studies in Svalbard, e.g., using an energy balance model (EBM) (Van Pelt et al., 2019) or statistical downscaling of a regional climate model (Noël et al., 2020) yield surface mass change components at 1 km or 500 m spatial resolution. In addition, the author claims that the impact of recent warming on the glacier mass balance in Svalbard and the Russian Arctic remains poorly known. While the Russian Arctic has not been extensively studied in the literature, previous works including, e.g., Lang et al. (2015), Van Pelt et al. (2019), Noël et al. (2020) discussed the recent mass change in Svalbard and associated driving processes. Furthermore, in a recent TC preprint, Sommer et al. (2020) discussed the recent mass change in the Russian Arctic, including the NZ and FJ regions. The author should consider mentioning these previous studies in the Introduction.

**Specific comments**

L15-16: The author should discuss how representative the selected northern glacier sites are with respect to the larger northern Svalbard region.

L16-18: While the reviewer agrees that climate models with grids ranging from 5-20 km spatial resolution may fail to represent small mountain glaciers, recent techniques including high-resolution EBM (Van Pelt et al., 2019) or statistical downscaling (Noël et al., 2020) allow refining climate model outputs to much higher spatial resolutions (sub-kilometer). The author could elaborate on this in L25-27. See also General comment #3.

L44-46: The authors could mention previous studies that discussed the drivers of recent mass change in Svalbard and Russian Arctic. See General comment #3.

L63: The concept of "Atlantification" is used here and in L296. Could the author clarify what is meant by "Atlantification" and refer the reader to a publication that further details this concept?

L102-103: How does the author distinguish between mountain glaciers and other ice masses in e.g. Svalbard? Could the author elaborate on the technique used.

L110: The author should mention that the selected mountain glaciers only cover ~2% of the total glacier area, and ~20% of the total mountain glacier area of the three studied regions. Are these selected glaciers representative of the whole mountain glacier area in SV, NZ and FJ? This issue should be discussed in the manuscript. See also General comment #2.

L159: The author could mention that spatial coverage can be as low as 23% (e.g. site 15, Table 1). What is the impact of very low spatial coverage on the resulting elevation change estimates? See General comment #1.

L178-180: The author could also use elevation changes (and uncertainties) in the different bins to estimate site-wide elevation change. Instead, the author used the site-wide volume change and glacier site area, why did the author choose to proceed this way? Does this affect the elevation change and uncertainty estimates?

L183: Does the author refer to the process of "firn compaction" when stating "cancels the dynamic component of elevation changes of glaciers."? Please, clarify.

L206-207: What is the impact of low spatial coverage on the elevation/mass change estimates? See also General comment #1.

L240-270: The author should consider mentioning the associated sites when referring to elevation changes across the Results section. For instance, in L229: "(from -1.58 m a$^{-1}$ at site 3 to -0.87 m a$^{-1}$ at site 5)". This holds for sections 4.1 to 4.3.

L246, L250 and 255: Fig. 4 should be split in two subpanels, i.e., a) Svalbard hypsometry and b) Russian Arctic hypsometry. The caption should be modified accordingly, and the author should refer to Fig. 4a after "ca. 900 m a.s.l.", "ca. 225-360 m a.s.l." and "ca. 480-600 m a.s.l.".

L247: Does the author mean -0.72 m a$^{-1}$ instead of -0.78 m a$^{-1}$ (see site 11 in Table 1)?

L263 and L270: The author should refer to Fig. 4b.

L305: Could the author add a regression line in Fig. 5 including SV-W, SV-C and SV-E glacier sites. Could the author mention the $R^2$ of the latter regression in the text?

L328-329: At sites 16 and 17, 57% and 62% of the glacier area is covered. It would be worth mentioning how this may impact the trends observed. For instance, if low-lying/rapidly thinning regions are not well captured, how does this translate in the estimated trends? In addition, Van Pelt et al. (2019) also found a positive surface mass balance trend in northern Svalbard.

L343-346: Is this statement supported by model data? Did the author compare the mass change from Noël et al. (2020) to their extrapolated estimates over mountain glaciers? Looking at figures in Noël et al. (2020) or Van Pelt et al. (2019), spatial patterns of surface mass balance at 500 m and 1 km resolution generally match quite well the elevation changes over the mountain glaciers shown by the author in the Supplementary Figures S30-S37. It is also important to note that all these techniques (including the current study) suffer from large (local) uncertainties. See also General comments #1 and #2. In brief, unless the author has formally evaluated/compared modelled data (e.g. from Noël et al. (2020) or Van Pelt et al. (2019)) against independent observations, L343-346 are insubstantial and could be removed.

L361-362: Are these two small glacier sites in North Svalbard representative of the larger northern ice-covered region? The author should further elaborate on this before drawing regional conclusions. See also General comment #2.

**Stylistic comments**

As a general comment, the author could consider using a capital letter for "Arctic" across the manuscript. The same comment holds for "Atlantic".

L77: The reviewer suggests: "air temperature in the period 2011-2017 was higher by … compared to the reference period 1981-2020 (Fig. 2a)."

L80: "whereas in September, it is most visible to the northeast of FJ (Fig. 2d)."

L106-107: The reviewer suggests: "were distinguished in SV (5), NZ (2) and FJ (2) respectively (Fig. 3). Overall, mountain glaciers were analysed at 29 sites: 19 in SV, 6 in NZ and 4 in FJ."

L117: "data suggested considerable differences."

L127: "these data suffer"

L131: "and incomplete temporal coverage"

L159: "part of the investigated glaciers"

L162: "glacier hypsometry, i.e., the area altitude distribution. To obtain … general glacier, 10-metre"

**Figures & Tables**

As a general comment, the author should consider adding extra tick marks on the color scales. This would enable a better visualization and interpretation of all figures.

Fig. 1d: Does the author mean "mm w.e. per year"?

Fig2: Sea ice extent is generally defined as the area showing > 15% of sea ice concentration, see for instance the sea ice products supplied by NSIDC.

**References**

Van Pelt et al. (2019): https://tc.copernicus.org/articles/13/2259/2019/

Lang et al. (2015): https://tc.copernicus.org/articles/9/83/2015/tc-9-83-2015.html

Noël et al. (2020): https://www.nature.com/articles/s41467-020-18356-1

Sommer et al. (2020): https://tc.copernicus.org/preprints/tc-2020-358/

---

## Referee Comment (RC2)

**Review of "Recent contrasting behaviour of mountain glaciers across the European High Arctic revealed by ArcticDEM data" by Jakub Małecki**

**General comments:**

The author uses the high-resolution elevation data from the ArcticDEM dataset to evaluate mass balance on selected glaciers in the European sector of the high Arctic, comprising Svalbard, Franz Josef Land, and Novaya Zemlya. The ice masses in these regions are among the "small" glaciers of the world (i.e. all the ice outside of the Greenland or Antarctic Ice Sheets), that make up relatively little of the overall global ice volume, but whose ongoing melting contributes a significant percentage of the current sea level rise. The analysis presented here restricts itself largely to the smallest of these "small" glaciers, namely land-terminating valley glaciers, with a few exceptions; it does not consider tidewater glaciers or ice caps.

**Specific comments:**

One aspect of satellite remote sensing is that the data are typically available to all; one can spend a great deal of time and energy analyzing data only to see another paper come out which supersedes one's own efforts. A recent paper by Hugonnet and others (2021) has seemingly eclipsed all other small glacier elevation change studies by marshalling all available elevation data (ASTER, ArcticDEM and REMA) in a major global analysis, including all glaciers in the three regions of the present study. Some mention is made of this, but it likely came midway during the writing of the present contribution. There are clear differences in the aims of the two studies, but ultimately this manuscript can seem a bit amputated in comparison. It is of course a more focused regional study, but it is not exhaustive since it does not analyze all the glaciers in these regions. It is higher-resolution, working at the original 2-m grid of the ADEMs trips, but this does not necessarily improve the results of the dH analysis, given the relatively large spatial coherence of that signal. Because of the many glitches in the ADEM data, the author has to hand-edit problem areas in the DEMs, even after filtering; this explains at least in part why the present study is restricted to a subset of the entire glacier-covered area, since hand-editing all glaciers would be very time-consuming.

By not considering tidewater glaciers, complications arising from having to account for the dynamic balance (frontal advance and retreat, calving) are side-stepped. Indeed, since surge-type glaciers tend to be mostly tidewater, this problem is also avoided. However, some mountain valley glacier *have* been observed to surge, on Svalbard at any rate, and surging is not mentioned at all. It is likely not an issue, since surging does not seem to generally occur in large clusters of neighboring valley glaciers in the same area.

One of the stated aims is to provide proxy information about the local climate, and in this regard, the paper contributes to our knowledge of these areas, particularly to less well-studied areas in the Russian Arctic. However, it would have been nice to see a more in-depth analysis of the connection between the derived glacier change and the results of mass balance models, rather than just trends in temperature and precipitation. The results of Noel et al. (2020) are considered, briefly, but those of van Pelt et al (2019) are not. The findings of the latter study need to be included, as they show positive balances in the same general area as in this study, in SV-N.

**Corrections:**

L3: Small land-terminating mountain glaciers are a widespread and important element of the Arctic, influencing local hydrology, microclimate, and ecosystems. Due to their relatively small ice volumes, this class of ice masses is particularly sensitive to the significant ongoing climate warming in the European sector…

L16: …experiences…

L17: …challenge for mass balance models…

L22: …has fundamentally improved our knowledge on the general trends of glacier change in the Arctic, and…

L25: However, these studies have relatively coarse spatial resolution, and thus more limited coverage of smaller glaciers.

L29: … class of glaciers north of the Arctic circle, and are vulnerable to climate warming due to their small ice volumes…

L30: …exhibit low…

L31: …and shallow ice thickness…

L31: …balance, being little modified…

L33: …might not only help predict future fluctuations…

L34: …piece…

L42: …fast rate, and that many…

L44: …disappear in a rapidly…

L45; …islands, and…

L53: The Barents Sea is strongly influenced by warm Atlantic waters (Fig. 1a), particularly to the west (e.g. West Spitsbergen Current) and south (e.g. North Cape Current , making the climate of these two sectors relatively mild (Figs. 1b and 1c) and wet (Fig. 1d). For this reason, waters west of SV and southwest of NZ are free of sea ice, even during the winter (Fig. 1e). During summer, the Barents Sea is typically ice-free, except for around FJ (Fig. 1f), where the Atlantic influence is reduced, making climate there the coolest and driest in the region.

L61: Over the past decades, the Barents Sea region has been experiencing strong ocean and atmospheric changes, the so-called Atlantification of the area (*appropriate reference*).

L67: …has been reported…

L75: The global climate reanalysis ERA5 dataset (Hersbach et al., 2020) shows that these trends prevail over the period of this study (roughly 2011-2017).

L77: …temperatures were…

L80: …and in September north and east of FJ (Fig. 2d). The ERA5 analysis…

L86: …approximately 85% are mountain glaciers, which comprise 14 % of the total ice area. Overall, glaciers of SV are known to have been losing mass over the past decades; the most recent estimate of the climatic mass balance is -8 Gt a-1, or ca. -0.23 m w.e. a-1, for the period 2000-2019 (Schuler et al., 2020). The larger glaciers of SV are typically polythermal, with a temperate base and cold surface layer, whereas the small glaciers…

L99: … that the small…

L103: … mountain glaciers, due to the paucity of studies of such glaciers.

L106: …distinguished (five, two and two, respectively for SV, NZ and FJ).

L109: … analysis, for consistency between the different regions.

L111: …NZ, and small…

L114: Therefore, the glacier…

L122: The basis for the glacier elevation change analysis is the high-resolution (2 m) digital elevation model (DEM) strip dataset downloaded from the ArcticDEM repository (https://www.pgc.umn.edu/data/arcticdem).

L125: …and have been used previously in various…

L127: However, the data suffer…

L128: Besides the usual issues common for glacier…

L130: …ArcticDEMs contain numerous artefacts

132: *Delete sentence* Solutions…section.

L137: …due to the scarcity…

L143: dimensions. To align the DEMs, I used the co-registration technique of  Nuth and Kääb (2011); however, this…

L149: moutonées

L155: Artefacts on glacier surfaces were manually removed by inspecting DEM hillshades for suspicious landforms (e.g. steep bumps, hollows, ridges or gullies on otherwise gentle glacier surfaces), and DoDs for unlikely patterns of glacier surface change (e.g. sharply defined areas of abnormally high or low *dh*). In some areas, the resulting data gaps were larger than areas with useful data, leaving some glacier DoDs with limited spatial coverage.

L165: … area, and the surface at an unknown time, but most likely…

L165: The mosaics were downsampled to 20 m with bilinear interpolation and corrected to orthometric heights using the EGM2008 geoid model (Pavlis et al., 2012), which is accurate to

within a few meters over the study regions, and is therefore sufficient for plotting the calculated glacier elevation changes against altitude.

L169: … fit to a 20-m grid of the…

L170: All glacier grid-points with dh data were exported with their…

L173 …percentiles, all of which…

L175: The hypsometry for each glacier was calculated from the glacier outlines and the downsampled and orthometrically corrected ArcticDEM mosaic data. Median values of $dh$ were calculated for each 50-m elevation bin, since the median is a more robust metric than the average, when there are outliers in the data

L183: … overall $\overline{dh}/dt$, since…

L184:  …advised caution when using such a conversion method…

L193: …sites, the empirical scaling of Martín-Español et al. (2015) was used, which is based on…

L198: …were summed to…

L204: …study, there is another important factor.

L216 …lowest or highest…

L216: …had insufficient…

L227: …(Fig. 3a), where all glaciers have been experiencing thinning, and at all elevations.

L229: …by the median…

Table 1: Date of first DEM, Date of last DEM

Table 1: *There is no apostrophe in decadal dates e.g. 2040s, rather than 2040's.*

L239: …the relatively low elevations of the mountain glaciers.

L240: From the available data, there is a northward trend of less negative elevations change.

L245: …the highest elevation ice masses…

L249: In SV-N, mountain glaciers have a contrasting pattern of change; here, glaciers have had positive…

L53: …between the rapid ice loss  of…

L254: …with a poleward…

L260: In the subregion NZ-N, all three sites of mountain glaciers showed glacier-wide thinning, with an…

L262: …NZ-S, had higher rates…

L263: …and rapid…

L265: …by low topography and extensive large ice caps, on most islands.

L275: …glaciers and high-elevation…

L281: *The Hugonnet study covers slightly different periods. Calculating trends with even a single year difference in the period can lead to very different results, if that year was remarkable in any way.*

L286: …while losses were…

L287: …values to the ~4600…

L288: …covered by…

L289: …regions; ca. 25%...

L294: …Atlantic…

L298: …SV, with two trends in thinning rates (Fig. 3a). There is an overall trend of less negative $\overline{dh/dt}$ to the north, and another to the interior of SV-C.

L303: …(Fig. S42), and close to zero at ca. 900 m a.s.l. This implies that over much of SV (SV-W, SV-C and SV- E), it is predominantly elevation that controls spatial variability…

L306: … slight trend of reducing thinning rates to the northeast (Figs. 3b and 3c).

L309: As many as 21 of the 29 sites investigated…

L310: …sites, *dh* was negative, even at…

L313: …and the dominant role of surface processes to their…

L314: …glaciers would eventually…

L316: …some sites had losses…

L327: …northeastern…

L339: …ice north of SV…

L340: An increase in snowfall is, however, not evident in the ERA5 reanalysis data (Fig. 2), which shows that while SV-N did experience strong sea ice retreat and a slight increase in winter precipitation, this increase was comparable to other subregions in which there was overall glacier mass loss, e.g. SV-E and NZ- N. Nor do the results of a finer-scale 500 m model by Noël et al. (2020) match the observations reported here, show instead a generally negative trend in B over Sites 16 and 17.

L344: *Higher resolution is not necessarily the answer. As mentioned in my general comments, the 1-km simulations of van Pelt et al. (2019) do hint at more positive balance in SV-N. The disagreement between model and observation can also lie in how the mass balance model is*

*constructed in the first place. Noël et al and van Pelt et al use regional climate model simulations for their forcing, both of which have resolutions of ca. ~10 km. These are then further downscaled in the mass balance model to the reported resolutions of 0.5 and 1 km respectively.*

L357: …even in the present…

L362: …part of SV) experienced…

L363: …the greatest climate warming…

L364: …interactions, even for low-activity ice masses, and…

L366: …this research provides a useful…

---

## Author Response (AR1)

Author's reply to the manuscript **"Recent contrasting behaviour of mountain glaciers across the European High Arctic revealed by ArcticDEM"**, by *J. Malecki* and submitted to *The Cryosphere*.

Dear Editor and Reviewers,
I am grateful for the time and instructive comments you made on my manuscript. Let me assure you I did my best to address all issues raised in your thorough reviews. I believe the corrected version will be closer to meeting the high standards of *The Cryosphere* journal.
Below you will find my responses to individual comments in blue, followed by a list of changes to the manuscript in red.
With kind regards,
Jakub Małecki

**Reply to the Reviewer 1**

**General comments**
1. While estimation of uncertainties is well described in the manuscript, uncertainties in elevation/volume/mass change can be large for different glacier sites. For instance, these uncertainties in elevation/mass balance can sometimes be equal to or double the absolute change for a given elevation bin (see Supplementary Tables). In addition, the spatial coverage of surveyed glacier can be as low as ~20%, with large data gap in given elevation bins (see the hypsometry graphs in the Supplementary Figures). The authors should elaborate on how these local uncertainties impact the results estimated for the different glacier sites, as well as how low spatial coverage affects the elevation change estimates. To address this issue, a sensitivity experiment exploring the impact of low spatial coverage on the elevation change estimates could be conducted by reducing the spatial coverage of a well-surveyed site (e.g. > 90% coverage). It is important to have more insight on how much spatial coverage of glacier sites is required to provide reliable estimates

The impact of high uncertainties in individual elevation bins is now clarified in Sect. 3.8. This impact is limited since high uncertainties typically refer to very small elevation bins, e.g. highest or lowest, comprising a very small proportion in the overall glacier area. A sensitivity test has also been performed and outlined in Section 3.8. The main finding is that cutting the data coverage from ~90% to ~20% changed the dh/dt only slightly, between 0.01 and 0.07 m/y (absolute values), depending on the site. A new figure has been added to the Methods section:

[Figure]

**Figure 3 Results of the sensitivity test performed on sites with data coverage > 90 %. Black bars – differences in the calculated glacier geodetic balances between the full-data coverage and reduced-data coverage; grey whiskers – uncertainty range of the reduced-data coverage balances; dashed whiskers – uncertainty range of the full-data coverage balances. For data coverage used in the test see Fig. S1.**

My interpretation on why the results are relatively little sensitive to a dramatic data removal is as follows: dh/dt of low-activity arctic glaciers is well-correlated with elevation due to the domination of surface processes

in their dh/dt. For this reason, a mosaic of small data patches, spread on several neighbouring glaciers, might not necessarily play a huge role in the final dh/dt estimate as long, as the largest elevation bins are captured.

2. The selected glacier sites only cover 1373 km2, which represents ~20% of the total mountain glacier area of the three regions (L287-288), and 2% of the total ice-covered area in SV, NZ and FJ. Out of these glacier sites, 804 km2 (59% of the studied area) can be used to estimate elevation/mass changes. The reviewer questions whether a 15-20% coverage of these mountain glaciers is sufficient to draw firm conclusions, notably the sharp regional contrast between site 1 and sites 16-17. For instance, are these two small glacier sites in North Svalbard representative of the surrounding northern region? The authors should elaborate on this topic.

For the data-coverage issue, please refer again to the sensitivity test, which provides an insight into how low coverage might potentially influence the calculated dh/dt. Over the study regions, SV, NZ and FJ, glacier responses appear not to change much over distances of ~50-100 km, except for north Svalbard (SV-N), so a denser grid of study sites might not necessarily bring much new information. The representativity of the selected study sites has been highlighted by comparisons of the observed glacier change patterns to the Hugonnet et al. (2021) global dataset (e.g. updated Fig. 4 with added dh/dt information from Hugonnet et al., 2020) and to the outcomes of climatic mass balance modelling studies (van Pelt et al., 2019; Noel et al., 2020).

3. Recent publications are omitted in the literature review, in particular the author claims that the spatial resolution in previous studies was not sufficient to provide reliable estimates of mass change over small mountain glaciers (L25-27). However previous studies in Svalbard, e.g., using an energy balance model (EBM) (Van Pelt et al., 2019) or statistical downscaling of a regional climate model (Noël et al., 2020) yield surface mass change components at 1 km or 500 m spatial resolution. In addition, the author claims that the impact of recent warming on the glacier mass balance in Svalbard and the Russian Arctic remains poorly known. While the Russian Arctic has not been extensively studied in the literature, previous works including, e.g., Lang et al. (2015), Van Pelt et al. (2019), Noël et al. (2020) discussed the recent mass change in Svalbard and associated driving processes. Furthermore, in a recent TC preprint, Sommer et al. (2020) discussed the recent mass change in the Russian Arctic, including the NZ and FJ regions. The author should consider mentioning these previous studies in the Introduction.

Thank you for this important point. This has been corrected in the revised version. I refer to the outcomes of high-resolution modelling by van Pelt et al. (2019) and Noel et al. (2020) across the manuscript. I also recalculated climatic balances over the study period with datasets provided with the aforementioned publications and compare these with my dh/dt observations. New Fig. 7 has been added to the discussion:

[Figure]

**Fig. 7 Example comparison of glacier mass balance models and elevation change observations at Site 16 in SV-N. Outcomes of climatic mass balance (CMB) models by (a) van Pelt et al. (2019) at 1000 m resolution, and (b) by Noël et al. (2020) at 500 m resolution. Observed $\overline{dh/dt}$ after (c) Hugonnet et al. (2021) at 100 m resolution, and (d) this study at 20 m resolution, downsampled from 2 m. Note the colour scale is common for CMB and $\overline{dh/dt}$. Hillshaded background: ArcticDEM, source: Polar Geospatial Center, University of Minnesota, Esri.**

The figure compares outcomes of mass balance models (van Pelt et al., 2019; Noel et al., 2020) and dh/dt observations by Hugonnet et al. (2021) and presented in this study. Some more maps are presented in new supplementary figures S44-S53.

**Specific comments**

L15-16: The author should discuss how representative the selected northern glacier sites are with respect to the larger northern Svalbard region.

I believe this has been improved, see my comment to the reviewer's general comment #2.

L16-18: While the reviewer agrees that climate models with grids ranging from 5-20 km spatial resolution may fail to represent small mountain glaciers, recent techniques including high-resolution EBM (Van Pelt et al., 2019) or statistical downscaling (Noël et al., 2020) allow refining climate model outputs to much higher spatial resolutions (sub-kilometer). The author could elaborate on this in L25-27. See also General comment #3.

This reviewer's comment refers to the sentence from the abstract: "The findings reveal that near-stagnant glaciers might exhibit contrasting behaviours (fast thinning vs. thickening) over relatively short distances, being a challenge for climate models, but also an opportunity to test their reliability". From the data presented in this study, it appears that only 1 of the 2 discussed hi-res CMB models was able to reproduce strong gradient between Site 1 (negative changes) and Site 16 (positive), some 30 km away, so I believe this situation might be considered challenging for CMB models. The progress in downscaling is undoubtful and the models by van Pelt

et al. (2019) and Noel et al. (2020) are discussed further in the Discussion section 5.3, rather than in the introduction.

L44-46: The authors could mention previous studies that discussed the drivers of recent mass change in Svalbard and Russian Arctic. See General comment #3.
Added.

L63: The concept of "Atlantification" is used here and in L296. Could the author clarify what is meant by "Atlantification" and refer the reader to a publication that further details this concept?
L102-103: How does the author distinguish between mountain glaciers and other ice masses in e.g. Svalbard? Could the author elaborate on the technique used.
Done, this is now clarified and backed up by proper citations.

L110: The author should mention that the selected mountain glaciers only cover ~2% of the total glacier area, and ~20% of the total mountain glacier area of the three studied regions. Are these selected glaciers representative of the whole mountain glacier area in SV, NZ and FJ? This issue should be discussed in the manuscript. See also General comment #2.
Corrected. Also, see the reply to general comment #2.

L159: The author could mention that spatial coverage can be as low as 23% (e.g. site 15, Table 1). What is the impact of very low spatial coverage on the resulting elevation change estimates? See General comment #1.
Corrected. Also, see the reply to general comment #1.

L178-180: The author could also use elevation changes (and uncertainties) in the different bins to estimate site-wide elevation change. Instead, the author used the site-wide volume change and glacier site area, why did the author choose to proceed this way? Does this affect the elevation change and uncertainty estimates?
I am not sure if I understand correctly. Perhaps the reviewer meant area-weighting of dh/dt obtained for individual elevation bins to obtain the site-wide dh/dt? If that was the case, such procedure is basically the same as summing all bin-specific volume change rates (since dv=dh x area) and dividing by the total glacier area.

L183: Does the author refer to the process of "firn compaction" when stating "cancels the dynamic component of elevation changes of glaciers."? Please, clarify.
Corrected, the 'dynamic component' referred to emergence/submergence.

L206-207: What is the impact of low spatial coverage on the elevation/mass change estimates? See also General comment #1.
Please, see the reply to general comment #1.

L240-270: The author should consider mentioning the associated sites when referring to elevation changes across the Results section. For instance, in L229: "(from -1.58 m a-1 at site 3 to -0.87 m a-1 at site 5)". This holds for sections 4.1 to 4.3.
Right, much better now.

L246, L250 and 255: Fig. 4 should be split in two subpanels, i.e., a) Svalbard hypsometry and b) Russian Arctic hypsometry. The caption should be modified accordingly, and the author should refer to Fig. 4a after "ca. 900 m a.s.l.", "ca. 225-360 m a.s.l." and "ca. 480-600 m a.s.l.".
As above.

L247: Does the author mean -0.72 m a-1 instead of -0.78 m a-1 (see site 11 in Table 1)?
Thanks, that was a typo. Corrected to -0.72.

L263 and L270: The author should refer to Fig. 4b.
Corrected.

L305: Could the author add a regression line in Fig. 5 including SV-W, SV-C and SV-E glacier sites. Could the author mention the R2 of the latter regression in the text?
Done. The linear regression for median elevation vs dh/dt, for SV-W, SV-C and SV-E, had r2=0.90.

L328-329: At sites 16 and 17, 57% and 62% of the glacier area is covered. It would be worth mentioning how this may impact the trends observed. For instance, if low-lying/rapidly thinning regions are not well captured, how does this translate in the estimated trends? In addition, Van Pelt et al. (2019) also found a positive surface mass balance trend in northern Svalbard.
Please refer to the reply to the general comment #1. dh/dt obtained even for data coverage of ~20% provides realistic estimates, although with much higher uncertainty range. Results by Van Pelt et al. (2019) have been cited in a number of places of the manuscript.

L343-346: Is this statement supported by model data? Did the author compare the mass change from Noël et al. (2020) to their extrapolated estimates over mountain glaciers? Looking at figures in Noël et al. (2020) or Van Pelt et al. (2019), spatial patterns of surface mass balance at 500 m and 1 km resolution generally match quite well the elevation changes over the mountain glaciers shown by the author in the Supplementary Figures S30-S37. It is also important to note that all these techniques (including the current study) suffer from large (local) uncertainties. See also General comments #1 and #2. In brief, unless the author has formally evaluated/compared modelled data (e.g. from Noël et al. (2020) or Van Pelt et al. (2019)) against independent observations, L343-346 are insubstantial and could be removed.
Comparisons of the observed elevation changes with the modelled mass balance data were performed on the recalculated data published alongside original papers. This part of the manuscript is now expanded in Sections 5.3 and 5.4 and in supplementary figs. S44-S51.

L361-362: Are these two small glacier sites in North Svalbard representative of the larger northern ice-covered region? The author should further elaborate on this before drawing regional conclusions. See also General comment #2.
Please refer to the reply to the general comment #2.

**Stylistic comments**
As a general comment, the author could consider using a capital letter for "Arctic" across the manuscript. The same comment holds for "Atlantic".
OK, done
L77: The reviewer suggests: "air temperature in the period 2011-2017 was higher by … compared to the reference period 1981-2020 (Fig. 2a)."
L80: "whereas in September, it is most visible to the northeast of FJ (Fig. 2d)."
L106-107: The reviewer suggests: "were distinguished in SV (5), NZ (2) and FJ (2) respectively (Fig. 3). Overall, mountain glaciers were analysed at 29 sites: 19 in SV, 6 in NZ and 4 in FJ."
L117: "data suggested considerable differences."
L127: "these data suffer"
L131: "and incomplete temporal coverage"
L159: "part of the investigated glaciers"
L162: "glacier hypsometry, i.e., the area altitude distribution. To obtain … general glacier, 10-metre"\
Thank you for the suggestions above, applied

**Figures & Tables**
As a general comment, the author should consider adding extra tick marks on the color scales. This would enable a better visualization and interpretation of all figures.
Yes, definitely right, corrected.

Fig. 1d: Does the author mean "mm w.e. per year"?
Yes, corrected.

Fig2: Sea ice extent is generally defined as the area showing > 15% of sea ice concentration, see for instance the sea ice products supplied by NSIDC.

Yes, but I still chose to show the 50% threshold in Fig. 2c and 2d since the shapes of 50% concentration lines nicely depict the ongoing sea ice changes over these regions.

**Reply to the Reviewer 2**

**Specific comments:**
One aspect of satellite remote sensing is that the data are typically available to all; one can spend a great deal of time and energy analyzing data only to see another paper come out which supersedes one's own efforts. A recent paper by Hugonnet and others (2021) has seemingly eclipsed all other small glacier elevation change studies by marshalling all available elevation data (ASTER, ArcticDEM and REMA) in a major global analysis, including all glaciers in the three regions of the present study. Some mention is made of this, but it likely came midway during the writing of the present contribution. There are clear differences in the aims of the two studies, but ultimately this manuscript can seem a bit amputated in comparison. It is of course a more focused regional study, but it is not exhaustive since it does not analyze all the glaciers in these regions. It is higher-resolution, working at the original 2-m grid of the ADEMs trips, but this does not necessarily improve the results of the dH analysis, given the relatively large spatial coherence of that signal. Because of the many glitches in the ADEM data, the author has to hand-edit problem areas in the DEMs, even after filtering; this explains at least in part why the present study is restricted to a subset of the entire glacier-covered area since hand-editing all glaciers would be very time-consuming.

While writing the first version of the manuscript, one of its aims was to bring new data on the state of small glaciers from understudied areas of the European Arctic. With the arrival of the Hugonnet et al. (2021) global dataset, i.e. in the revised version presented here, I tried to slightly shift the centre of gravity of the manuscript, so that it focuses more on the practical potential of the presented data – for validation/calibration of mass balance models or for comparison of the performance of larger-scale mass balance assessments (van Pelt et al., 2019; Noel et al., 2020; Hugonnet et al., 2021).

By not considering tidewater glaciers, complications arising from having to account for the dynamic balance (frontal advance and retreat, calving) are side-stepped. Indeed, since surge-type glaciers tend to be mostly tidewater, this problem is also avoided. However, some mountain valley glacier *have* been observed to surge, on Svalbard at any rate, and surging is not mentioned at all. It is likely not an issue, since surging does not seem to generally occur in large clusters of neighboring valley glaciers in the same area.

Thank you for pointing that out, this is now corrected briefly. In fact, arctic surge-type mountain glaciers in the quiescent phase might be even better for extracting the mass balance signal from their elevation changes since these might display even lower horizontal and vertical velocities than normal glaciers. A small subset of the study glaciers is known/suspected to be of surge-type, but to my knowledge these have not been surging over many (> 5) past decades.

One of the stated aims is to provide proxy information about the local climate, and in this regard, the paper contributes to our knowledge of these areas, particularly to less well-studied areas in the Russian Arctic. However, it would have been nice to see a more in-depth analysis of the connection between the derived glacier change and the results of mass balance models, rather than just trends in temperature and precipitation. The results of Noel et al. (2020) are considered, briefly, but those of van Pelt et al (2019) are not. The findings of the latter study need to be included, as they show positive balances in the same general area as in this study, in SV-N.

The revised version refers to the results of van Pelt and Noel in a number of places. The comparison of these models with the observed changes is briefly discussed in Sect. 5.3 and shown in new Fig.7 and new supplementary figures S44-S53, so that the volume of the manuscript does not go out of control. Also, please refer to my response to the general comment #3 by Reviewer 1.

**Corrections:**

L3: Small land-terminating mountain glaciers are a widespread and important element of the Arctic, influencing local hydrology, microclimate, and ecosystems. Due to their relatively small ice volumes, this class of ice masses is particularly sensitive to the significant ongoing climate warming in the European sector…

L16: …experiences…

L17: …challenge for mass balance models…

L22: …has fundamentally improved our knowledge on the general trends of glacier change in the Arctic, and…

L25: However, these studies have relatively coarse spatial resolution, and thus more limited coverage of smaller glaciers.

L29: … class of glaciers north of the Arctic circle, and are vulnerable to climate warming due to their small ice volumes…

L30: …exhibit low…

L31: …and shallow ice thickness…

L31: …balance, being little modified…

L33: …might not only help predict future fluctuations…

L34: …piece…

L42: …fast rate, and that many…

L44: …disappear in a rapidly…

L45; …islands, and…

L53: The Barents Sea is strongly influenced by warm Atlantic waters (Fig. 1a), particularly to the west (e.g. West Spitsbergen Current) and south (e.g. North Cape Current , making the climate of these two sectors relatively mild (Figs. 1b and 1c) and wet (Fig. 1d). For this reason, waters west of SV and southwest of NZ are free of sea ice, even during the winter (Fig. 1e). During summer, the Barents Sea is typically ice-free, except for around FJ (Fig. 1f), where the Atlantic influence is reduced, making climate there the coolest and driest in the region.

L61: Over the past decades, the Barents Sea region has been experiencing strong ocean and atmospheric changes, the so-called Atlantification of the area (*appropriate reference*).

L67: …has been reported…

L75: The global climate reanalysis ERA5 dataset (Hersbach et al., 2020) shows that these trends prevail over the period of this study (roughly 2011-2017).

L77: …temperatures were…

L80: …and in September north and east of FJ (Fig. 2d). The ERA5 analysis…

L86: …approximately 85% are mountain glaciers, which comprise 14 % of the total ice area. Overall, glaciers of SV are known to have been losing mass over the past decades; the most recent estimate of the climatic mass balance is -8 Gt a-1, or ca. -0.23 m w.e. a-1, for the period 2000-2019 (Schuler et al., 2020). The larger glaciers of SV are typically polythermal, with a temperate base and cold surface layer, whereas the small glaciers…

L99: … that the small…

L103: … mountain glaciers, due to the paucity of studies of such glaciers.

L106: …distinguished (five, two and two, respectively for SV, NZ and FJ).

L109: … analysis, for consistency between the different regions.

L111: …NZ, and small…

L114: Therefore, the glacier…

L122: The basis for the glacier elevation change analysis is the high-resolution (2 m) digital elevation model (DEM) strip dataset downloaded from the ArcticDEM repository (https://www.pgc.umn.edu/data/arcticdem).

L125: …and have been used previously in various…

L127: However, the data suffer…

L128: Besides the usual issues common for glacier…

L130: …ArcticDEMs contain numerous artefacts

132: *Delete sentence* Solutions…section.

L137: …due to the scarcity…

L143: dimensions. To align the DEMs, I used the co-registration technique of Nuth and Kääb (2011); however, this…

L149: moutonées

L155: Artefacts on glacier surfaces were manually removed by inspecting DEM hillshades for suspicious landforms (e.g. steep bumps, hollows, ridges or gullies on otherwise gentle glacier surfaces), and DoDs for unlikely patterns of glacier surface change (e.g. sharply defined areas of abnormally high or low *dh*). In some

areas, the resulting data gaps were larger than areas with useful data, leaving some glacier DoDs with limited spatial coverage.

L165: … area, and the surface at an unknown time, but most likely…

L165: The mosaics were downsampled to 20 m with bilinear interpolation and corrected to orthometric heights using the EGM2008 geoid model (Pavlis et al., 2012), which is accurate to within a few meters over the study regions, and is therefore sufficient for plotting the calculated glacier elevation changes against altitude.

L169: … fit to a 20-m grid of the…

L170: All glacier grid-points with dh data were exported with their…

L173 …percentiles, all of which…

L175: The hypsometry for each glacier was calculated from the glacier outlines and the downsampled and orthometrically corrected ArcticDEM mosaic data. Median values of *dh* were calculated for each 50-m elevation bin, since the median is a more robust metric than the average, when there are outliers in the data

L183: … overall $d\bar{\bar{h}}/dt$, since…

L184: …advised caution when using such a conversion method…

L193: …sites, the empirical scaling of Martín-Español et al. (2015) was used, which is based on…

L198: …were summed to…

L204: …study, there is another important factor.

L216 …lowest or highest…

L216: …had insufficient…

L227: …(Fig. 3a), where all glaciers have been experiencing thinning, and at all elevations.

L229: …by the median…

Table 1: Date of first DEM, Date of last DEM

Table 1: *There is no apostrophe in decadal dates e.g. 2040s, rather than 2040's.*

L239: …the relatively low elevations of the mountain glaciers.

L240: From the available data, there is a northward trend of less negative elevations change.

L245: …the highest elevation ice masses…

L249: In SV-N, mountain glaciers have a contrasting pattern of change; here, glaciers have had positive…

L53: …between the rapid ice loss of…

L254: …with a poleward…

L260: In the subregion NZ-N, all three sites of mountain glaciers showed glacier-wide thinning, with an…

L262: …NZ-S, had higher rates…

L263: …and rapid…

L265: …by low topography and extensive large ice caps, on most islands.

L275: …glaciers and high-elevation…

Thank you so much, all of the corrections above have been introduced.

L281: *The Hugonnet study covers slightly different periods. Calculating trends with even a single year difference in the period can lead to very different results, if that year was remarkable in any way.*
Yes, I agree. I changed the sentence and mention different periods of observation in the first place, as a potential cause for the discrepancy between Hugonnet et al. (2021) and my data.

L286: …while losses were…

L287: …values to the ~4600…

L288: …covered by…

L289: …regions; ca. 25%...

L294: …Atlantic…

L298: …SV, with two trends in thinning rates (Fig. 3a). There is an overall trend of less negative $d\bar{\bar{h}}/dt$ to the north, and another to the interior of SV-C.

L303: …(Fig. S42), and close to zero at ca. 900 m a.s.l. This implies that over much of SV (SV-W, SV-C and SV- E), it is predominantly elevation that controls spatial variability…

L306: … slight trend of reducing thinning rates to the northeast (Figs. 3b and 3c).

L309: As many as 21 of the 29 sites investigated…

L310: …sites, *dh* was negative, even at…

L313: …and the dominant role of surface processes to their…

L314: …glaciers would eventually…

L316: …some sites had losses…

L327: …northeastern…
L339: …ice north of SV…
L340: An increase in snowfall is, however, not evident in the ERA5 reanalysis data (Fig. 2), which shows that while SV-N did experience strong sea ice retreat and a slight increase in winter precipitation, this increase was comparable to other subregions in which there was overall glacier mass loss, e.g. SV-E and NZ- N. Nor do the results of a finer-scale 500 m model by Noël et al. (2020) match the observations reported here, show instead a generally negative trend in B over Sites 16 and 17.
Thank you so much, all of the corrections above have been introduced.

L344: *Higher resolution is not necessarily the answer. As mentioned in my general comments, the 1-km simulations of van Pelt et al. (2019) do hint at more positive balance in SV-N. The disagreement between model and observation can also lie in how the mass balance model is constructed in the first place. Noël et al and van Pelt et al use regional climate model simulations for their forcing, both of which have resolutions of ca. ~10 km. These are then further downscaled in the mass balance model to the reported resolutions of 0.5 and 1 km respectively.*
This is true, I deleted the sentence on the need for even higher resolution simulations.

L357: …even in the present…
L362: …part of SV) experienced…
L363: …the greatest climate warming…
L364: …interactions, even for low-activity ice masses, and…
L366: …this research provides a useful…

Thank you so much, all of the corrections above have been introduced.

**LIST OF THE MOST IMPORTANT CHANGES:**

1 INTRODUCTION

- Changed one of the aims of the study

2 STUDY AREA

- Added ticks to scales in Fig. 1 and 2
- Added information about glacier surges in Svalbard and Novaya Zemlya

3 METHODS

- Added information about the impact of high dh/dt uncertainties of single elevation bins on the overall dh/dt of a given study site
- Performed a sensitivity test to get more information on how low data coverage might influence site-wide dh/dt measurements of glaciers
- Added a new figure (Fig.3) showing the results of the sensitivity test

4 RESULTS

- Changed the key figure (Fig. 4) by adding information about dh/dt of all mountain glaciers in SV, NZ and FJ based on RGI v6.0 and the dataset by Hugonnet et al. (2021)
- Corrected Figs. 5 and 6 according to the reviewers' suggestions

5 DISCUSSION

- Moved the subsection "Comparison with other studies" from the first to the third position (from 5.1 to 5.3)
- Section 5.3 has been expanded with a paragraph about the outcomes of previous glacier mass balance modelling in Svalbard
- New Fig. 7 added – comparison of simulated/observed changes
- Extended the discussion in Section 5.4

SUPPLEMENT

- Added new Figs. S1 and S44-S53
- Changed Fig. S43
- Changed Tabs. S1-S30 from the PDF format to the XLSX format to make their use easier

SUPPLEMENT

- Added new Figs. S1 and S44-S53
- Changed Fig. S43
- Changed Tabs. S1-S30 from the PDF format to the XLSX format to make their use easier

---

## Author Response (AR2)

Dear Editors and the Reviewer,

Thank you very much for your time spent on my manuscript "Recent contrasting behaviour of mountain glaciers across the European High Arctic revealed by ArcticDEM data". I am very happy to receive your positive feedback on the corrected version. I have now applied all suggestions by the Reviewer, i.e.:

L205 – I added a sentence "This implies that the procedure does not account for firn compaction, a process that might contribute to a slight surface lowering in high-elevation areas of some glaciers."

L327 – I added information about the slope of the regression (median elevation vs. geodetic balance), equal to 0.25 m a$^{-1}$/100 m.

I also corrected the four stylistic issues and changed the colour of fjords/ocean on detailed maps (Fig. 7 and supplementary figures) from blue to grey, so that water is not confused with positive mass balance areas.

Thanks again for your work.

Sincerely,

Jakub Małecki